# Know More, Know Clearer: A Meta-Cognitive Framework for Knowledge Augmentation in Large Language Models

**Hao Chen** [1]  **Ye He** [1]  **Yuchun Fan** [2]  **Yukun Yan** [3]  **Zhenghao Liu** [2]  **Qingfu Zhu** [1]  **Maosong Sun** [3]  **Wanxiang Che** [1]

## Abstract

Knowledge augmentation has significantly enhanced the performance of Large Language Models (LLMs) in knowledge-intensive tasks. However, existing methods typically operate on the simplistic premise that model performance equates with internal knowledge, overlooking the knowledge-confidence gaps that lead to overconfident errors or uncertain truths. To bridge this gap, we propose a novel meta-cognitive framework for reliable knowledge augmentation via differentiated intervention and alignment. Our approach leverages internal cognitive signals to partition the knowledge space into mastered, confused, and missing regions, guiding targeted knowledge expansion. Furthermore, we introduce a cognitive consistency mechanism to synchronize subjective certainty with objective accuracy, ensuring calibrated knowledge boundaries. Extensive experiments demonstrate the our framework consistently outperforms strong baselines, validating its rationality in not only enhancing knowledge capabilities but also fostering cognitive behaviors that better distinguish knowns from unknowns. All codes are available at `https://github.com/AI9Stars/Know-More-Know-Clearer`.

> *"To know that we know what we know, and to know that we do not know what we do not know, that is true knowledge."*
>
> — *Nicolaus Copernicus*

## 1. Introduction

Large Language Models (LLMs) have achieved remarkable proficiency through knowledge augmentation (Singh et al., 2025; Guo et al., 2025). However, existing knowledge augmentation methods typically operate on the simplistic premise that model performance is implicitly equated with internal knowledge (Andriopoulos & Pouwelse, 2023; Wang et al., 2025a; Chen et al., 2025a), where errors are equated with knowledge deficits and correct responses with mastery (Zhang et al., 2024). This perspective overlooks the intricate internal states underlying these outcomes (Mallen et al., 2023a). In practice, LLMs frequently assign high confidence to incorrect predictions triggering hallucinations (Huang et al., 2025), or exhibit unwarranted uncertainty even when generating accurate responses (Chhikara, 2025). Such phenomena reveal a fundamental misalignment between a model's subjective cognition and its objective correctness (Manggala et al., 2024). Consequently, blindly supplementing models with external knowledge fails to resolve the underlying cognitive misalignment and may even exacerbate it (Wen et al.; Wang et al., 2025c).

To address this misalignment, we first examine whether internal signals can reliably indicate model's underlying knowledge state (Liu et al., 2025). Through large-scale statistical analysis, we observe a stable exponential-like decay between prediction correctness and uncertainty (Figure 1), revealing a latent alignment between confidence-related signals and potential performance. This suggests that confidence reflects intrinsic knowledge states instead of merely arising as a byproduct of decoding (Bentegeac et al., 2025). This insight motivates a meta-cognitive perspective that reinterprets model failures as misalignments between confidence and accuracy, rather than mere knowledge deficits.

Building on a meta-cognitive perspective (Didolkar et al., 2024), we propose a knowledge augmentation framework that replaces indiscriminate supplementation with differentiated interventions, and leverage internal state modeling to systematically correct cognitive misalignment. Specifically, our framework comprises two core components: (i) *Cognition-Guided Knowledge Expansion (Know More)*. We enable the model to autonomously partition its knowledge space into mastered, confused, and missing regions, facilitating targeted supervision to augment knowledge where most needed. (ii) *Cognition-Driven Knowledge Calibration (Know Clearer)*. We introduce a cognitive consistency alignment mechanism to synchronize subjective confidence

---

[1]Harbin Institute of Technology [2]Northeastern University [3]Tsinghua University. Correspondence to: Yukun Yan <yanyk.thu@gmail.com>, Wanxiang Che <car@ir.hit.edu.cn>.

*Proceedings of the 43rd International Conference on Machine Learning*, Seoul, South Korea. PMLR 306, 2026. Copyright 2026 by the author(s).

with objective accuracy, ensuring the model's cognition is precisely calibrated to its knowledge capacity.

Extensive experiments across diverse QA tasks demonstrate that our framework significantly outperforms strong baselines, achieving competitive SOTA performance. Moreover, superior results on self-knowledge benchmarks demonstrate that our framework significantly enhances cognitive capabilities, enabling precise decision-making and clearer knowledge boundaries. Further analyses reveal that the correlation between confidence and accuracy becomes significantly more pronounced, indicating that internal representations are maturing into states of higher cognitive consistency. These results suggest that knowledge augmentation is not merely a matter of acquiring more knowledge, but of knowing more clearly what they know and what they do not.

## 2. Preliminaries

In this section, we introduce the conceptual foundations and empirical laws of meta-cognition, followed by the formal definition of Group Relative Policy Optimization (GRPO).

### 2.1. Meta-Cognition: Foundations and Empirical Laws

**Conceptual Foundations of Meta-cognition.** Meta-cognition, a concept pioneered by Flavell, is defined as "cognition about cognition", encompassing two critical dimensions: *monitoring* (assessing one's own knowledge accuracy) and *control* (regulating behavior based on such assessments). This framework was originally developed to characterize human cognitive development, yet its core principles map naturally onto the behavioral patterns observed in LLMs. In LLMs, this duality translates into a mechanism of internal governance: *monitoring* enables an introspective scrutiny of latent cognitive states, while *control* facilitates the self-regulatory adjustment of cognitive processes (Ji-An et al., 2025). Ideally, this loop allows the model to identify its understanding boundaries and ensure that external behaviors remain congruent with internal cognitive capacities.

**Discovery of the Structural Decay Law.** To quantify meta-cognitive capacity, we examine the relationship between internal signals and empirical performance. We define the Uncertainty ($\mathcal{U}$) as the sequence-level average negative log-likelihood (NLL) (Aichberger et al., 2026). For a generated reasoning path $y = \{x_1, x_2, \ldots, x_T\}$, $\mathcal{U}$ is formulated as:

$$\mathcal{U}(y|q) = -\frac{1}{T}\sum_{t=1}^{T}\log p_\theta(x_t|q, x_{<t}).\qquad(1)$$

By fitting mean accuracy across uncertainty intervals over large-scale samples, we observe a stable and universal regularity, as illustrated in Figure 1. Specifically, the empirical mean Accuracy ($Acc$) exhibits a consistent exponential de-

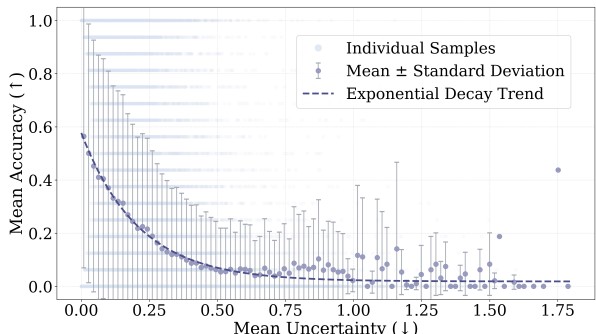

*Figure 1.* The Structural Decay Law between Accuracy and Uncertainty in Qwen2.5-7B-Instruct across Various QA Tasks.

cay relative to the uncertainty:

$$\mathbb{E}[Acc \mid \mathcal{U}] \approx a \cdot \exp(-\mathcal{U}) + b.\qquad(2)$$

We refer to this phenomenon as the "Structural Decay Law", its implementation and universal validation across diverse architectures are detailed in Appendix C.1. This regularåity suggests internal confidence signals carry structured information aligned with performance, not random fluctuations.

**Confidence as an Externalized Manifestation.** Although derived from raw distributions, $\mathcal{U}$ functions as a confidence-related heuristic for navigating the knowledge state. The Structural Decay Law reveals that empirical reliability is the inverse manifestation of $\mathcal{U}$: as $\mathcal{U}$ diminishes, implicit "answerability" increases predictably. From this perspective, confidence is an externalized reflection of this internal state, mirroring estimated success. This perspective aligns with meta-cognition: the stable mapping along the curve represents effective monitoring, whereas as uncertainty escalates, deviations from this curve signify an alignment failure where internal signals no longer accurately index reliability.

**Summary and Implications.** In summary, the Structural Decay Law provides a structural basis for our meta-cognitive framework. Building on this, we propose a framework operationalizing both meta-cognitive dimensions: **monitoring** to perceive internal knowledge states for targeted augmentation, and **control** to calibrate the gap between internal confidence and actual performance.

### 2.2. Group Relative Policy Optimization (GRPO)

To operationalize meta-cognitive optimization, we employ GRPO (Guo et al., 2025), which optimizes the policy by comparing a group of $G$ sampled outputs $\{y^{(1)}, \ldots, y^{(G)}\}$ against their group-relative advantage. For a query $q$, the training objective $\mathcal{L}_{\text{pg}}$ is defined as:

$$\mathcal{L}_{\text{pg}}(\theta) = -\frac{1}{G}\sum_{k=1}^{G}\min\left(r_\theta^{(k)}\mathbb{A}^{(k)}, \text{clip}\left(r_\theta^{(k)}, 1-\epsilon, 1+\epsilon\right)\mathbb{A}^{(k)}\right),$$
$$(3)$$

where $r_\theta^{(k)} = \frac{\pi_\theta(y^{(k)}|q)}{\pi_{\theta_{\text{old}}}(y^{(k)}|q)}$ is the importance weight, and $\mathbb{A}^{(k)}$ represents group-normalized relative advantages:

$$\mathbb{A}^{(k)} = \frac{r^{(k)} - \text{mean}(\{r^{(i)}\})}{\text{std}(\{r^{(i)}\})}. \tag{4}$$

Additionally, a KL divergence penalty $\mathcal{L}_{\text{KL}} = \mathbb{D}_{\text{KL}}(\pi_\theta \| \pi_{\text{ref}})$ is incorporated to maintain training stability.

## 3. Method

This section details the two core components of our framework: Cognition-Guided Knowledge Expansion (§3.1) and Cognition-Driven Knowledge Calibration (§3.2).

### 3.1. Cognition-Guided Knowledge Expansion (CGKE)

To achieve *Know More*, we formalize knowledge augmentation as a state-dependent injection process. Rather than treating all model failures as uniform knowledge deficiency, we diagnosticly demarcate the model's knowledge space into three distinct regions based on instance-level internal states. This allows for tailored expansion strategies specifically targeted to the model's current knowledge boundaries.

**Meta-Cognitive Region Assignment.** To operationalize cognitive region assignment, we probe the model's internal state for a given query $q$ and its gold answer $a$ via stochastic decoding. By sampling $K$ independent responses $\{\hat{a}^{(k)}\}_{k=1}^K$ from the model $\mathcal{M}$, we construct a behavior profile:

$$\mathcal{B}(q, a) = \left\{ \left( \hat{a}^{(k)}, u^{(k)}, \ell^{(k)} \right) \right\}_{k=1}^K, \tag{5}$$

where $u^{(k)}$ denotes the internal uncertainty (Eq. 1) and $\ell^{(k)}$ is the correctness of the $k$-th sample. We then implement the assignment mapping $\Gamma(\mathcal{B}) \rightarrow r \in \{\text{Mastered, Confused, Missing}\}$ by leveraging the model's intrinsic evaluative capacity via a meta-cognitive region assignment prompt (Appendix B). Grounded in a *performance-oriented* principle (Chen et al., 2024), this mapping synthesizes mean accuracy and mean uncertainty as primary reference indicators to partition the instance space: Mastered instances exhibit robust internal representations with high stability and minimal uncertainty, while Missing instances reflect a fundamental knowledge void through stochastic patterns and elevated uncertainty. Confused instances manifest fragmented performance and fragile reasoning, indicating partial knowledge without meta-cognitive alignment.

**Region-Specific Data Augmentation.** Building on these assigned knowledge states, we implement a targeted augmentation pipeline $\Phi(q, a, r)$ that transforms each instance into specialized training samples $(q', a')$. For any given instance, we first identify *Cognitive Tags*, which comprise the critical entities, logical prerequisites, or core concepts that act as the primary subjects of the identified knowledge state. These tags serve as the semantic anchors for formulating a high-precision search query used to retrieve external grounding passages $\mathcal{P}$. Conditioned on the assigned region $r$, the data is reconstructed as follows:

- **Knowledge Missing (Epistemic Foundation)**: For instances lacking foundational representations, the augmentation focuses on knowledge instantiation. We treat the retrieved passages $\mathcal{P}$ as the primary authoritative source to synthesize training samples targeting missing definitions and core factual pillars. This process ensures the model learns to map previously unrecognized queries to the concrete evidence contained within $\mathcal{P}$, effectively filling cognitive blind spots.

- **Knowledge Confused (Structural Disambiguation)**: For instances of fragmented or unstable understanding, we leverage the structural information and relational constraints within $\mathcal{P}$ to enforce representational alignment. By refining queries into clarified formulations and decomposed sub-questions, we anchor the reasoning process to explicit evidence, thereby resolving entity ambiguity and ensuring latent knowledge is correctly activated according to the grounding context.

- **Knowledge Mastered (Boundary Expansion)**: For instances demonstrating robust consistency, we pursue knowledge extensibility by leveraging $\mathcal{P}$ to identify cross-concept comparisons or broader contextual connections. Beyond mere factual replication, this encourages the model to synthesize new insights from retrieved evidence, extending its stable knowledge boundary while preserving representational integrity.

**Unified Supervised Fine-Tuning Objective.** Guided by these region-specific strategies, we define a unified optimization objective. For a training distribution $\mathcal{D}$, we optimize the model on the cognitively-transformed instances $(q', a')$:

$$\mathcal{L}_{\text{sft}} = \mathbb{E}(q'_r, a'_r) \sim \mathcal{D} \left[ \ell(q'_r, a'_r) \right], \tag{6}$$

where $\ell(\cdot)$ is the standard supervised loss. This stage ensures that the model acquires a more robust and expansive knowledge base (*Know More*), providing a solid foundation for the knowledge calibration (*Know Clearer*).

### 3.2. Cognition-Driven Knowledge Calibration (CDKC)

While Cognition-Guided Knowledge Expansion enables the model to structure knowledge (*Know More*), it lacks explicit regularization for aligning internal confidence with empirical performance. Consequently, models may still exhibit cognitive misalignment, especially with unfamiliar knowledge. To bridge this gap, we introduce Cognition-Driven

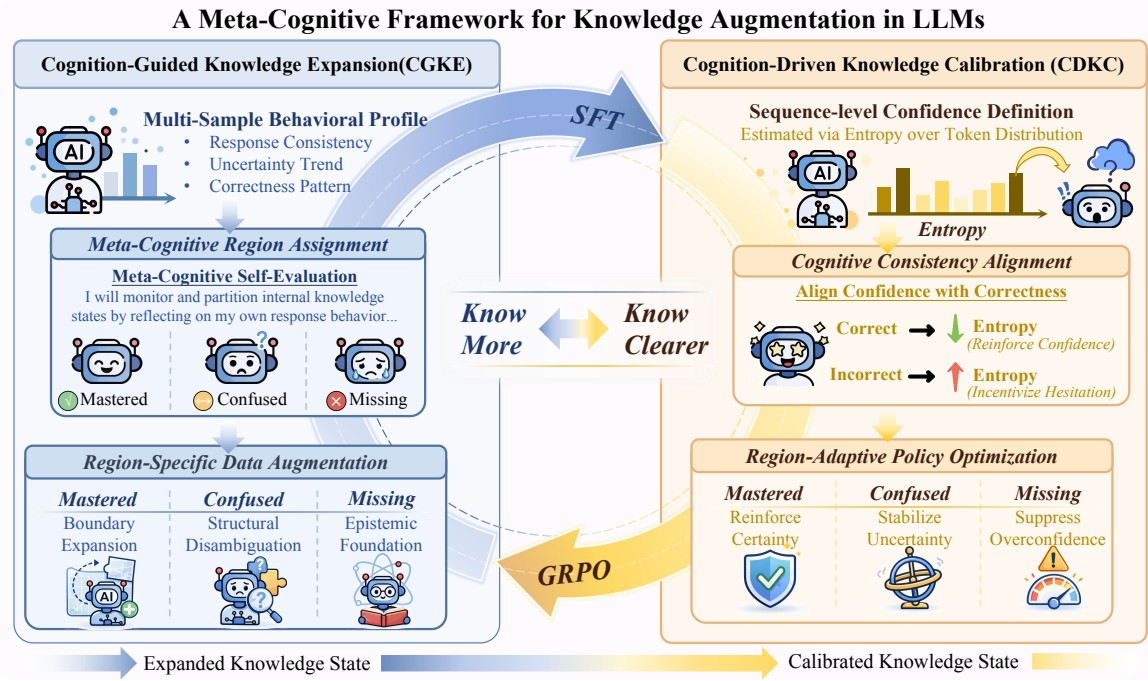

*Figure 2.* Overview of the Meta-Cognitive Knowledge Augmentation Framework. The Cognition-Guided Knowledge Expansion (CGKE) module enables differentiated knowledge augmentation guided by internal cognitive signals. The Cognition-Driven Knowledge Calibration (CDKC) module calibrates subjective confidence with objective correctness, promoting clearer cognitive knowledge boundaries.

Knowledge Calibration, which reshapes the model's confidence landscape to achieve the objective of *Know Clearer*.

**Sequence-level Confidence Definition.** To formalize the measurement of internal uncertainty, we define the sequence-level entropy $\mathcal{H}$ (Shannon, 1948) of a reasoning path $y^{(k)}$ as distribution-aware confidence metric. For a query $q$, $\mathcal{H}$ captures the average probabilistic dispersion of the model's policy $\pi_\theta$ across the entire generation process (Kadavath et al., 2022), thereby quantifying the degree of predictive uncertainty inherent in the sequence:

$$\mathcal{H}(\pi_\theta \mid q, y^{(k)}) = -\frac{1}{T}\sum_{t=1}^{T}\sum_{v\in\mathcal{V}}\pi_\theta(v \mid q, y^{(k)}_{<t}) \tag{7}$$
$$\cdot \log \pi_\theta(v \mid q, y^{(k)}_{<t}),$$

where $\mathcal{V}$ is the vocabulary, $T$ is the sequence length, and $y^{(k)}_{<t}$ represents the prefix tokens.

**Cognitive Consistency Alignment.** Leveraging these uncertainty signals, we propose Cognitive Consistency Alignment to actively recalibrate the model's confidence. Rather than applying a uniform entropy penalty, we modulate the model's confidence distribution via a binary meta-cognitive alignment objective. This approach employs a reward signal $S_\theta$ calculated based on the accuracy of each reasoning path to serve as a directional switch for entropy regularization. Accordingly, we recalibrate the model's probabilistic dispersion in accordance with its empirical competence. The

calibration loss is formulated as:

$$\mathcal{L}_{\mathrm{cal}}(\theta) = \mathbb{E}_{q\sim\mathcal{D}, \{y^{(k)}\}\sim\pi_\theta}\left[\frac{1}{G}\sum_{k=1}^{G}\alpha^{(k)}\cdot\mathcal{H}(\pi_\theta \mid q, y^{(k)})\right], \tag{8}$$

where $G$ denotes the number of sampled reasoning paths per query, $\alpha^{(k)}$ acts as a performance-contingent multiplier:

$$\alpha^{(k)} = \begin{cases} +1, & \text{if } \mathcal{S}_\theta > 0 \quad \textit{(Reinforce Confidence)} \\ -1, & \text{if } \mathcal{S}_\theta \leq 0 \quad \textit{(Incentivize Hesitation)}. \end{cases} \tag{9}$$

Specifically, for correct reasoning paths, this mechanism constrains the distribution to be peaked to *reinforce confidence*; for incorrect paths, it pushes the distribution toward uniformity to *incentivize hesitation*.

**Region-Adaptive Policy Optimization.** The final objective integrates the calibration signal into the GRPO framework. To ensure the optimization is sensitive to the model's underlying knowledge state, we combine the standard policy gradient with our cognitive consistency loss. The total training objective is formulated as:

$$\mathcal{L}_{\mathrm{total}} = \mathcal{L}_{\mathrm{pg}} + \lambda_1\mathcal{L}_{\mathrm{KL}} + \lambda_2\mathcal{L}_{\mathrm{cal}}(\theta), \tag{10}$$

where $\mathcal{L}_{\mathrm{pg}}$ and $\mathcal{L}_{\mathrm{KL}}$ are defined as in Section 2.2. The coefficients $\lambda_1$ and $\lambda_2$ modulate the strength of distribution stability and cognitive regularization, respectively.

This objective inherently accommodates the model's state by tailoring regularization across heterogeneous knowledge regions. In *knowledge missing* regions, the loss primarily suppresses overconfident hallucinations by rewarding high entropy; in *knowledge confused* regions, it stabilizes the confidence landscape across competing hypotheses; and in *knowledge mastered* regions, it reinforces cognitive resolve.

# 4. Experimental Methodology

This section details our experimental setup, covering datasets, metrics, baselines, and implementation details. **Dataset.** Our approach is evaluated across various commonsense and complex multi-hop QA benchmarks. Based on the model's inherent proficiency, we categorize these into three Knowledge Grounding States: (1) **Weakly-Grounded**: PopQA (Mallen et al., 2023b), Musique (Trivedi et al., 2022), and SQuAD (Rajpurkar et al., 2016); (2) **Partially-Grounded**: NQ (Kwiatkowski et al., 2019), HotpotQA (Yang et al., 2018), 2WikiMQA (Ho et al., 2020), BeerQA (Qi et al., 2021), WebQuestions (Berant et al., 2013), and Bamboogle (Press et al., 2023); (3) **Well-Grounded**: SearchQA (Dunn et al., 2017) and TriviaQA (Joshi et al., 2017). Specifically, NQ, HotpotQA and 2WikiMQA serve as in-domain data for training-phase, while others are used for out-of-domain evaluation.

**Evaluation Metrics.** We adopt accuracy as the primary evaluation metric across all QA tasks, following previous work (Li et al., 2024b; Chen et al., 2025a).

**Baselines.** For comprehensive comparison, we adopt three representative paradigms following the trajectory of "knowing more" and "knowing clearer": (1) **Fundamental Capabilities**: *Vanilla LLM* to assess the model's inherent parametric knowledge; *Chain-of-Thought (CoT)* (Wei et al., 2022) to evaluate internal reasoning through the elicitation of intermediate logical steps; *Retrieval-Augmented Generation (RAG)* (Lewis et al., 2020) augments LLMs by retrieving relevant external knowledge and incorporating them into the generation process. (2) **Knowledge Expansion (Know More)**: *Vanilla SFT* (Roberts et al., 2020) to perform standard supervised fine-tuning on open-source corpora; *LLKD-SFT* (Li et al., 2025a) to implement confidence-related knowledge distillation from a corresponding model series with tenfold larger parameters; and our *CGKE* to expand knowledge boundaries based on meta-cognitive feedback. (3) **Knowledge Calibration (Know Clearer)**: *Know What* (Kapoor et al., 2024) to elicit uncertainty awareness through supervised fine-tuning with correctness labels; *CRew-DPO* (Du et al., 2025) to enhance performance by leveraging model confidence as reward signals for preference optimization; *BARREL* (Yang et al., 2025) to develop boundary awareness via GRPO, enabling the model to withhold responses and express uncertainty when appropriate;

*GRPO* (Guo et al., 2025) serves as the foundational reinforcement learning baseline for policy alignment by computing rewards relative to group-sampled outputs; and our *CDKC* to calibrate the alignment between internal confidence and actual performance through meta-cognitive optimization. To ensure a fair comparison, all methods in the calibration stage are consistently implemented based on the CGKE-enhanced model.

**Implementation Details.** In our experiments, we adopt Qwen2.5-7B-Instruct (Qwen et al., 2025) and Llama-3.1-8B-Instruct (Grattafiori et al., 2024) as backbone models. For retrieval module, we employ the UltraRAG (Chen et al., 2025b) framework with Qwen3-Embedding-0.6B (Zhang et al., 2025) to retrieve the top-5 contexts from the Wikipedia corpus. For the SFT phase, we follow the Wu et al. and utilize the TRL (von Werra et al., 2020) framework for 1 epoch with a learning rate of $5 \times 10^{-5}$, employing LoRA (Hu et al., 2022) for parameter-efficient fine-tuning. For the GRPO phase, we utilize the verl (Sheng et al., 2025) framework for 1 epoch with a learning rate of $1 \times 10^{-6}$. The reward coefficients $\lambda_1$ and $\lambda_2$ are both set to 0.001, where $\lambda_1$ follows the standard KL coefficient setting (Guo et al., 2025) and $\lambda_2$ is set equal to $\lambda_1$ to ensure balanced regularization strength. All experiments are conducted on four NVIDIA A100-80GB GPUs. More experimental details are provided in Appendix B for completeness.

# 5. Results and Analysis

In this section, we first evaluate the overall performance of our framework and conduct an ablation study to assess the impact of each component. Furthermore, we assess model self-knowledge, present a visualization of calibration reliability, and analyze the evolution of Structural Decay Law to illuminate the meta-cognitive alignment process. More complete results are provided in Appendix C.

## 5.1. Overall Performance

We evaluate the performance of our framework on 11 QA benchmarks under varying knowledge grounding states for the Qwen2.5-7B-Instruct and Llama-3.1-8B-Instruct models in Table 1. Overall, our proposed framework significantly outperforms all baseline methods, achieving substantial gains across diverse knowledge benchmarks.

**Effectiveness of Knowledge Expansion and Calibration.** In the knowledge expansion stage, compared with undifferentiated fine-tuning paradigms, our CGKE achieves the most significant performance gains. By leveraging meta-cognitive feedback, it targets diverse knowledge requirements and effectively extends the boundaries of the model's parametric knowledge. The knowledge calibration stage further elevates performance by refining the model's meta-cognitive

*Table 1.* Overall Performance. The highest scores are emphasized in **bold**, while the second highest scores are marked with an underline. Methods in the Know More phase are based on the *vanilla LLM*, whereas those in the Know Clearer phase are based on our *CGKE*.

| Method | Weakly-Grounded | | | Partially-Grounded | | | | | | Well-Grounded | | AVG |
|---|---|---|---|---|---|---|---|---|---|---|---|---|
| | PopQA | MusQ | SQuAD | NQ | HotQA | 2Wiki | BeerQA | WebQ | Bamboo | SeaQA | TriQA | |
| *Qwen2.5-7B-Instruct* | | | | | | | | | | | | |
| *Fundamental Capabilities* | | | | | | | | | | | | |
| Vanilla LLM | 14.50 | 2.69 | 14.67 | 20.63 | 22.47 | 27.70 | 21.23 | 31.74 | 20.80 | 51.80 | 55.13 | 25.76 |
| CoT (2022) | 15.07 | 7.16 | 16.67 | 23.60 | 27.00 | 33.47 | 23.77 | 32.48 | 37.60 | 58.27 | 59.30 | 30.40 |
| RAG (2020) | **43.44** | 6.87 | **35.13** | 37.30 | 38.73 | 33.80 | 39.10 | 33.61 | 19.20 | 54.13 | 66.90 | 37.11 |
| *Knowledge Expansion (Know More)* | | | | | | | | | | | | |
| Vanilla SFT (2020) | 15.90 | 5.42 | 16.17 | 22.50 | 25.23 | 28.83 | 23.23 | 33.17 | 23.20 | 58.43 | 55.73 | 27.98 |
| LLKD-SFT (2025a) | 15.43 | 4.80 | 15.20 | 22.43 | 25.57 | 35.73 | 22.97 | 33.96 | 27.20 | 57.07 | 54.80 | 28.65 |
| CGKE (ours) | 16.13 | 5.79 | 15.77 | 22.53 | 26.30 | 36.90 | 24.37 | 33.51 | 24.80 | 57.90 | 56.67 | 29.15 |
| *Knowledge Calibration (Know Clearer)* | | | | | | | | | | | | |
| Know What (2024) | 16.53 | 6.21 | 15.70 | 22.63 | 26.17 | 39.33 | 23.47 | 33.51 | 23.20 | 56.13 | 54.77 | 28.88 |
| CRew-DPO (2025) | 16.57 | 5.54 | 16.13 | 22.73 | 27.67 | 42.80 | 24.90 | 34.01 | 26.40 | 60.00 | 56.10 | 30.26 |
| BARREL (2025) | 20.97 | 4.14 | 15.87 | 24.87 | 30.43 | 49.87 | 26.00 | 38.44 | 24.80 | 61.97 | 66.07 | 33.04 |
| GRPO (2025) | 28.10 | 14.19 | 29.23 | 37.13 | 42.13 | 58.77 | 39.10 | 47.79 | 33.60 | 77.83 | 75.13 | 43.91 |
| CDKC (ours) | 28.27 | 15.64 | 31.13 | 38.57 | 42.97 | 59.53 | 39.63 | 47.64 | 36.80 | 78.40 | 76.10 | 44.97 |
| CDKC (w/ 2 round) | 31.67 | **18.66** | 33.70 | **42.70** | **46.07** | **60.87** | **42.97** | **52.17** | **39.20** | **84.07** | **79.13** | **48.29** |
| *Llama-3.1-8B-Instruct* | | | | | | | | | | | | |
| *Fundamental Capabilities* | | | | | | | | | | | | |
| Vanilla LLM | 20.33 | 4.47 | 15.23 | 31.03 | 25.70 | 32.13 | 23.50 | 34.35 | 24.80 | 63.00 | 65.50 | 30.91 |
| CoT (2022) | 24.33 | 8.44 | 17.37 | 34.03 | 32.10 | 33.90 | 25.53 | 36.91 | 40.80 | 66.07 | 70.67 | 35.47 |
| RAG (2020) | 42.73 | 6.25 | **35.07** | 38.20 | 40.23 | 35.53 | 40.10 | 32.19 | 19.20 | 55.10 | 70.20 | 37.71 |
| *Knowledge Expansion (Know More)* | | | | | | | | | | | | |
| Vanilla SFT (2020) | 29.43 | 7.70 | 19.00 | 34.47 | 31.43 | 42.00 | 26.40 | 39.67 | 21.60 | 62.07 | 67.77 | 34.69 |
| LLKD-SFT (2025a) | 25.07 | 7.07 | 17.43 | 33.73 | 31.07 | 43.40 | 25.60 | 38.34 | 20.00 | 68.90 | 69.23 | 34.44 |
| CGKE (ours) | 26.60 | 8.23 | 19.63 | 36.67 | 34.90 | 47.53 | 29.23 | 41.78 | 22.40 | 66.47 | 68.27 | 36.52 |
| *Knowledge Calibration (Know Clearer)* | | | | | | | | | | | | |
| Know What (2024) | 26.97 | 8.23 | 18.33 | 32.97 | 32.77 | 46.70 | 28.10 | 39.52 | 24.00 | 65.37 | 68.40 | 35.58 |
| CRew-DPO (2025) | 28.97 | 10.14 | 21.53 | 35.20 | 38.83 | 50.60 | 31.83 | 42.42 | 24.00 | 68.33 | 72.30 | 38.56 |
| BARREL (2025) | 35.00 | 10.80 | 28.20 | 40.73 | 43.90 | 56.43 | 37.67 | 49.95 | 35.20 | 82.90 | 81.43 | 45.66 |
| GRPO (2025) | 37.33 | 18.58 | 30.37 | 47.70 | 48.20 | 60.70 | 40.70 | 50.98 | 45.60 | 84.47 | 82.23 | 49.71 |
| CDKC (ours) | 40.33 | 20.40 | 31.67 | 49.83 | 49.67 | 61.77 | 42.13 | 53.84 | 48.00 | 84.83 | 83.33 | 51.44 |
| CDKC (w/ 2 round) | **43.93** | **26.02** | 34.70 | **53.93** | **53.80** | **63.10** | **45.23** | **55.91** | **60.00** | **88.23** | **85.63** | **55.50** |

clarity. Specifically, CDKC consistently outperforms competitive alignment baselines, including uncertainty-aware methods like BARREL and the policy optimization methods like GRPO. It delivers average improvements of 1.06% on Qwen2.5-7B-Instruct and 1.73% on Llama-3.1-8B-Instruct over the strongest baseline. These results reveal that calibrating the alignment between internal confidence and actual performance is crucial for reliable knowledge elicitation.

**Superiority of Iterative Evolution.** Notably, CDKC (w/ 2 round) yields the most substantial gains by completing two full "expansion-calibration" cycles, which represents the iterative evolution of our framework. It delivers an additional 3.32% improvement on Qwen2.5-7B and 4.06% on Llama-3.1 over the single-round CDKC. This creates a reciprocal loop where first round calibrated uncertainty guides targeted knowledge expansion, providing higher-quality seeds for subsequent calibration. Such progressive refinement is particularly evident in complex tasks like MusQ, which demand precise knowledge grounding.

**Robustness across Grounding States.** Our framework demonstrates robustness across all grounding states. In well-grounded scenarios, CDKC (w/ 2 round) attains near-saturated performance by fully eliciting inherent expertise. Most notably, CDKC achieves superior performance compared to RAG without introducing any additional retrieval overhead, and consistently outperforms CoT-based approaches while requiring far fewer tokens, demonstrating that effective performance gains can be realized at no extra inference cost. Such gains across the grounding spectrum underscore our framework's superiority, regardless of initial knowledge density.

*Table 2.* Ablation Study on Qwen2.5-7B-Instruct across Diverse QA Tasks.

| Method | Weakly-Grounded | | | Partially-Grounded | | | | | | Well-Grounded | | AVG |
|---|---|---|---|---|---|---|---|---|---|---|---|---|
| | PopQA | MusQ | SQuAD | NQ | HotQA | 2Wiki | BeerQA | WebQ | Bamboo | SeaQA | TriQA | |
| *Knowledge Expansion (Know More)* | | | | | | | | | | | | |
| CGKE | 16.13 | **5.79** | 15.77 | **22.53** | **26.30** | **36.90** | **24.37** | 33.51 | 24.80 | 57.90 | **56.67** | **29.15** |
| w/o Knowledge Mastered | 16.10 | **5.79** | 15.67 | **22.53** | 25.93 | 35.53 | 24.27 | **33.61** | **25.60** | **57.97** | 56.40 | 29.04 |
| w/o Knowledge Confused | 16.00 | 5.05 | **15.80** | 22.40 | 25.47 | 30.23 | 23.80 | 33.56 | 24.80 | 57.40 | 56.07 | 28.23 |
| w/o Knowledge Missing | **16.57** | 5.50 | 15.63 | 22.23 | 25.30 | 31.97 | 24.27 | 33.27 | 24.00 | 57.27 | 56.13 | 28.38 |
| *Knowledge Calibration (Know Clearer)* | | | | | | | | | | | | |
| CDKC | **28.27** | **15.64** | **31.13** | **38.57** | **42.97** | **59.53** | **39.63** | 47.64 | **36.80** | **78.40** | **76.10** | **44.97** |
| w/o CGKE | 26.30 | 12.99 | 28.50 | 35.53 | 42.07 | 58.20 | 38.23 | 45.77 | 32.00 | 76.87 | 75.03 | 42.86 |
| w/o Cognitive Alignment | 28.10 | 14.19 | 29.23 | 37.13 | 42.13 | 58.77 | 39.10 | **47.79** | 33.60 | 77.83 | 75.13 | 43.91 |

## 5.2. Ablation Study

To further dissect the contribution of each component, we conduct ablation experiments as summarized in Table 2.

**Impact of Targeted Knowledge Expansion.** We first examine the Knowledge Expansion (Know More) stage to assess how different knowledge categories contribute to parametric growth. In this stage, removing any specific knowledge category among the Mastered, Confused, and Missing subsets consistently degrades performance. Notably, the absence of "Knowledge Confused" leads to the most pronounced decline, with the average score dropping from 29.15 to 28.23. This highlights that resolving cognitive ambiguity is more impactful for expanding knowledge boundaries than simply reinforcing mastered facts.

**Necessity of Knowledge Calibration.** Regarding the Knowledge Calibration (Know Clearer) stage, removing Cognitive-Guided Knowledge Expansion (CGKE) leads to a noticeable performance decline, underscoring that a pre-expanded knowledge base serves as a vital foundation for effective alignment. Furthermore, the drop without Cognitive Alignment proves that knowledge alone is insufficient. Explicit calibration is essential to elicit reliable knowledge and mitigate hallucinations across all grounding levels.

## 5.3. Meta-cognition Assessment in Self-Knowledge Task

To evaluate the meta-cognition of LLMs, we conduct experiments on a self-knowledge benchmark comprising answerable and unanswerable questions (Yin et al., 2023), designed to evaluate the model's cognitive alignment and its ability to identify knowledge boundaries. During the evaluation, the model is encouraged to output "the answer is unknown" when it perceives a question as beyond its knowledge.

**Cognitive Behavior Analysis.** To quantify this process, we define four cognitive states based on question type and decision correctness, as illustrated in Figure 3. For answerable questions, *True Positives (TP)* represent correct cognitive decisions to provide an answer, while *False Negatives*

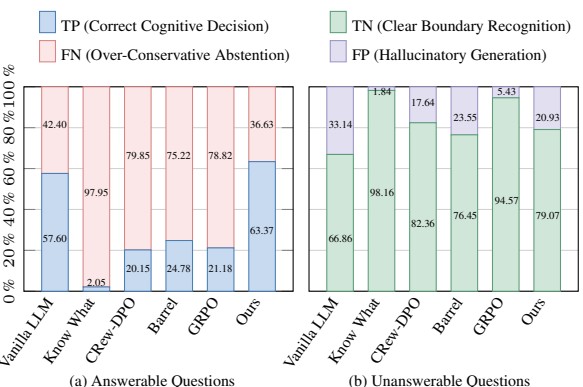

(a) Answerable Questions (b) Unanswerable Questions

*Figure 3.* Distribution of Cognitive Decision States. Comparison on (a) answerable questions and (b) unanswerable questions across different methods based on Qwen2.5-7B-Instruct backbone.

*(FN)* denote over-conservative refusal of known topics. For unanswerable questions, *True Negatives (TN)* reflect boundary recognition by correctly identifying ignorance, whereas *False Positives (FP)* signify hallucinatory generation.

On answerable questions, our framework achieves the most significant proficiency with a 63.37% TP rate, whereas baselines like "Know What" exhibit extreme over-conservatism with a 97.95% FN rate, failing to elicit latent knowledge. On unanswerable questions, while baselines like "Know What" reach a higher refusal accuracy of 98.16% TN, their poor performance on answerable tasks indicates indiscriminate rejection rather than true self-knowledge. In contrast, our framework achieves a 12.21% increase in boundary recognition (TN) over the vanilla model while maintaining a dominant 63.37% TP rate. This demonstrates a unique ability to align internal certainty with objective correctness, effectively avoiding biased conservative strategies. We provide a theoretical analysis of this phenomenon in Appendix A.

**Multi-dimensional Evaluation.** To provide a more intuitive understanding of the model's cognitive performance, we report the actual accuracy on answerable questions (Figure 4,

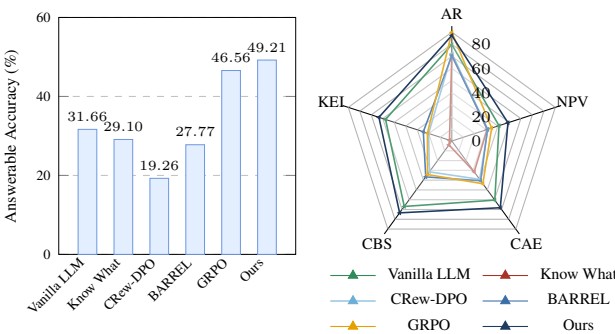

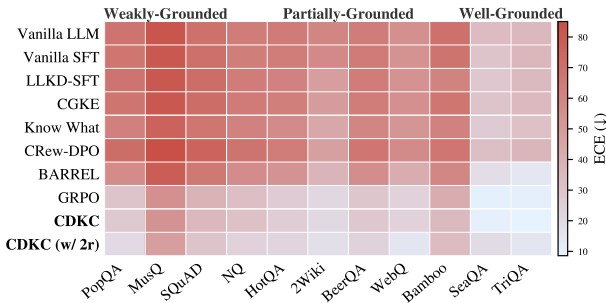

*Figure 4.* Performance Comparison across Different Metrics on the Qwen2.5-7B-Instruct. Left: Answerable accuracy. Right: Multidimensional evaluation on AR, KEI, CBS, CAE, and NPV metrics.

*Figure 5.* Visualization of ECE Performance across Different Methods and Datasets on the Qwen2.5-7B-Instruct Backbone. Lower ECE (lighter color) indicates better calibration.

left) alongside multi-dimensional cognitive metrics (Figure 4, right): Answer Reliability (AR), Negative Predictive Value (NPV), Knowledge Elicitation Index (KEI), Cognitive Balance Score (CBS), and Cognitive Alignment Efficiency (CAE), with formal definitions provided in Appendix C.2.

On answerable questions, our framework achieves the most significant accuracy, which is closely tied to its superior cognitive decision-making. Multi-dimensional evaluation confirms our framework's dominance across nearly all metrics, proving its reliability in both knowledge elicitation (KEI) and boundary recognition (NPV). This superior meta-cognitive equilibrium synchronizes internal certainty with objective veracity, effectively navigating the trade-off between over-conservatism and hallucination.

## 5.4. Visualization of Calibration Reliability via Expected Calibration Error (ECE)

In this section, we visualize the calibration effectiveness of our approach by analyzing the Expected Calibration Error (ECE) (Kapoor et al., 2024) across various knowledge-grounded tasks. To quantify the alignment between the model's subjective confidence and its objective accuracy, we calculate ECE by partitioning predictions into $M = 10$ equal-mass bins based on their confidence scores. The confidence for each sample is derived from its average uncertainty using the mapping $c = e^{-NLL}$. For each bin $B_m$, we compute the absolute difference between the average confidence and the actual accuracy, with the final ECE being the weighted sum of these gaps across all bins:

$$ECE = \sum_{m=1}^{M} \frac{|B_m|}{N} |acc(B_m) - conf(B_m)|. \quad (11)$$

This metric provides a rigorous measure of whether the model is overconfident or well-calibrated when deploying knowledge, where a lower ECE value indicates superior calibration performance.

As illustrated in Figure 5, the heatmap reveals consistent patterns along both axes. Along the vertical axis, vanilla and SFT-based methods exhibit persistently deep colors, indicating severe overconfidence across all datasets. In contrast, CDKC and CDKC (w/ 2 round) concentrate the lightest colors, demonstrating a substantial reduction in calibration error and confirming that our method successfully rectifies the model's internal uncertainty estimation, ensuring its self-reported confidence aligns more closely with its actual performance. Along the horizontal axis, a gradual transition from deeper to lighter colors is observed as the grounding condition shifts from Weakly-Grounded to Well-Grounded. This pattern reflects an intuitive phenomenon: models tend to express higher and more calibrated confidence on knowledge they have internalized more thoroughly, while remaining more uncertain on sparsely grounded content. Taken together, these two perspectives suggest that CDKC not only improves overall calibration, but also preserves and enhances the model's intrinsic sensitivity to its own knowledge boundaries. Complete experimental results are provided in Appendix C.5.

## 5.5. Analysis of the Optimized Structural Decay Law

In this section, we examine the optimized structural decay law introduced in Chapter 2, focusing on the correlation between uncertainty-accuracy alongside density distributions.

On Qwen2.5-7B-Instruct, CDKC fundamentally transforms the joint distribution of uncertainty and accuracy. As shown in Figure 6a, the base model exhibits a noisy exponential decay pattern, where samples are scattered irregularly across the uncertainty-accuracy space, reflecting an unstable and unpredictable calibration state. Following optimization, this disordered distribution converges toward a structured and monotonic decay, with the calibration curve becoming markedly smoother and more predictable. Concretely, the density of high-accuracy samples in the low-uncertainty region increases substantially, while the proportion of low-accuracy samples is effectively com-

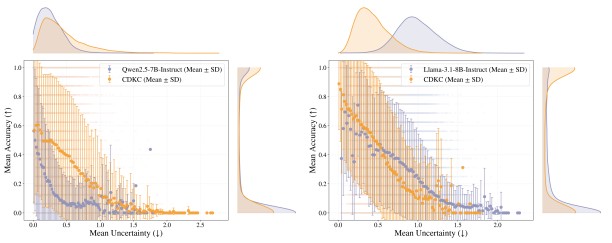

*(a)* Qwen2.5-7B-Instruct    *(b)* Llama-3.1-8B-Instruct

*Figure 6.* Visualization of the Structural Decay Law before and after optimization with CDKC on Qwen2.5-7B-Instruct and Llama-3.1-8B-Instruct backbones.

pressed, shifting the overall prediction distribution toward a higher-quality regime. Furthermore, the overconfidence phenomenon is significantly suppressed: samples that previously exhibited low uncertainty despite low accuracy are notably reduced, and the model's uncertainty estimates for challenging samples tend toward a more conservative and epistemically appropriate range. Collectively, these changes indicate that CDKC enables the model's internal uncertainty to serve as a high-fidelity proxy for its actual performance, establishing a more reliable cognitive boundary.

Similarly, for Llama-3.1-8B-Instruct (Figure 6b), the sample accuracy distribution exhibits a consistent trend with the observations on Qwen2.5-7B-Instruct, further validating the generalizability of our approach across architectures. Notably, the base Llama model displays a pronounced over-conservative tendency, where a substantial proportion of samples cluster in the high-uncertainty region despite possessing non-trivial accuracy, reflecting an under-confident cognitive state that fails to leverage the model's actual capabilities. CDKC effectively rectifies this pathology by pulling these dispersed and conservative cognitive states into a concentrated predictive interval, substantially reducing unnecessary abstention while preserving appropriate caution on genuinely difficult samples. Beyond merely shifting scores, this optimization establishes a stronger negative correlation between uncertainty and accuracy. This confirms that CDKC synchronizes subjective confidence with objective veracity across architectures, rather than promoting blind boldness or excessive caution.

## 6. Related work

**Meta-Cognition in LLMs.** Recently, meta-cognition has been introduced into LLMs to reflect models' self-awareness (Wang et al., 2025b). Initial efforts (Wang & Zhao, 2024) explore that meta-cognitive prompting can effectively enhance reasoning performance during inference. Subsequently, some research (Ma et al., 2025; Wang et al., 2025b) demonstrates that internal representations naturally encode meta-cognitive signals. These insights have cat-

alyzed a shift toward adaptive inference regulation (Sui et al., 2025; Zhao et al., 2026) and strategic knowledge coordination. For instance, some RAG systems (Zhou et al., 2024; Mombaerts et al., 2024; Yuan et al., 2025) integrate meta-cognitive feedback to rectify retrieval failures and optimize knowledge acquisition strategies, while MeCo (Li et al., 2025c) introduces adaptive triggers for tool invocation. Joishy et al. further leverage meta-cognitive intervention to resolve knowledge conflicts. While these works validate meta-cognition in reasoning, they rely on inherent signals rather than explicit optimization.

**Knowledge augmentation and calibration.** Knowledge augmentation for LLMs is typically categorized as parametric or non-parametric paradigms. Non-parametric methods like RAG (Lewis et al., 2020; Fan et al., 2024) integrate external knowledge during inference by organizing and synthesizing retrieved content (Wang et al., 2024). Conversely, parametric approaches internalize knowledge into weights via distillation (Xu et al., 2024) or continual learning (Roberts et al., 2020), typically utilizing SFT (Ovadia et al., 2024) for parameter updates (Huang et al., 2026). Despite their progress, ensuring reliability remains challenging (Fan et al., 2025a), as it requires a cognitive grasp of the model's own knowledge (Zheng et al., 2025; Fan et al., 2025b). Representative methods quantify uncertainty (Liu et al., 2025) from internal signals to derive confidence-based rewards (Geng et al., 2024; Du et al., 2025), thereby penalizing over-confident yet error outputs. Beyond probability alignment, research explores uncertainty expression and abstention (Kapoor et al., 2024; Li et al., 2024a), empowering models to refuse answering based on their perceived knowledge limits. Other works discuss boundary recognition to help models distinguish between known and unknown internal knowledge (Yang et al., 2025; Li et al., 2025b). Unlike existing methods, we introduce meta-cognitive principles to guide knowledge expansion and calibration, offering a new perspective on knowledge-augmented reliability.

## 7. Conclusion

This paper presents a novel meta-cognitive framework for reliable knowledge augmentation via differentiated intervention and alignment. We discover a universal exponential decay relationship between uncertainty and accuracy, providing a theoretical foundation for meta-cognitive alignment in LLMs. Building on this insight, we partition the knowledge space into mastered, confused, and missing regions to enable targeted intervention, and introduce a bidirectional entropy-based optimization algorithm for systematic confidence calibration. Experimental results demonstrate that our framework significantly elevates knowledge accuracy and optimizes cognitive capabilities, leading to clearer knowledge boundaries and structured internal states.

## Acknowledgments

We gratefully acknowledge the support of the National Natural Science Foundation of China (NSFC) via grant 62236004 and 62476073. We also gratefully acknowledge the support of the AI9Stars community for their valuable contributions to this research.

## Impact Statement

This work does not need ethical considerations, as it only utilizes open-source foundation models and publicly available datasets. This paper presents work whose goal is to advance the field of Machine Learning. There are many potential societal consequences of our work, none of which we feel must be specifically highlighted here.

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

# A. Theoretical Analysis of Meta-cognitive Calibration

In this section, we analyze knowledge augmentation as a process of dynamic manifold reshaping. While GRPO acts as a stochastic amplifier for latent reasoning paths, reward sparsity often triggers a "KL-Refusal Trap", causing the policy manifold to collapse into conservative, non-functional regions. To counteract this, we propose Cognition-Driven Knowledge Calibration (CDKC). By injecting absolute meta-cognitive signals, CDKC functions as a corrective vector field that repairs the manifold: employing centripetal forces to reinforce correct reasoning and centrifugal forces to dissolve defensive attractors. This mechanism ensures the model evolves into a metacognitively aware reasoner rather than a passive reward-seeker.

## A.1. GRPO as a Cognitive Path Amplifier: A Stochastic Dynamics Perspective

The Group Relative Policy Optimization (GRPO) framework optimizes the policy $\pi_\theta$ by estimating the advantage through intra-group reward comparison. The gradient of the objective $\mathcal{J}_{GRPO}(\theta)$ with respect to parameters $\theta$ is formulated as:

$$\nabla_\theta \mathcal{J}_{GRPO}(\theta) \approx \frac{1}{G} \sum_{i=1}^{G} \underbrace{\left( \frac{R(q, a_i) - \mu_G}{\sigma_G} \right)}_{\text{Advantage } \hat{A}_i} \cdot \underbrace{\frac{\pi\theta(a_i|q)}{\pi_{old}(a_i|q)}}_{r\theta^{(i)}} \cdot \underbrace{\nabla_\theta \log \pi_\theta(a_i|q)}_{\text{Directional Guide}}, \tag{12}$$

where $G$ denotes the group size, while $\mu_G$ and $\sigma_G$ represent the mean and standard deviation of rewards within the group, respectively. While Eq. (3) utilizes a clipped surrogate objective to ensure stability, Eq. (12) captures the first-order optimization dynamics. At the onset of each update where $\pi\theta \approx \pi_{old}$ and $r_\theta^{(i)} \to 1$, the gradient direction aligns with the standard policy gradient. From a cognitive dynamics perspective, GRPO functions as a *path amplifier* that refines the model's internal logic through two synergistic processes:

**Mechanism of Logical Discovery via Positive Feedback**     For a given query $q$, GRPO establishes a dynamic mechanism for identifying and elevating superior reasoning trajectories. In scenarios where a subset of sampled trajectories $\mathcal{A}^+ = \{a_i \mid R(q, a_i) > \mu_G\}$ represents the model's discovery of a valid logical path, the Advantage Factor $\hat{A}_i$ becomes positive. This triggers a constructive feedback loop: the gradient flow, aligned with the Directional Guide $\nabla_\theta \log \pi_\theta$, actively shifts the probability mass of the policy toward the parameter regions defined by $\mathcal{A}^+$. By rapidly amplifying the likelihood of these high-reward paths, the framework converts sporadic cognitive breakthroughs into stable policy updates, effectively isolating coherent logical signals from the high-dimensional noise of the pre-trained prior.

**Cognitive Survival of the Fittest: Selective Pressure on Reasoning Paths**     Beyond simple probability adjustment, GRPO facilitates a competitive selection process akin to "survival of the fittest" within the latent representation space. By utilizing the intra-group mean $\mu_G$ as a baseline and $\sigma_G$ as a measure of logical consistency, the framework identifies and reinforces subtle representation differences that lead to correct outcomes. This mechanism filters out flawed heuristics and rewards trajectories that exhibit robust causal reasoning. Effectively expanding the model's problem-solving upper bound during the early stages of training, this process anchors correct logical chains within the policy manifold, transforming stochastic sampling into structured, high-order cognitive capabilities.

## A.2. Limitations of Native GRPO: Amplification of Conservatism Bias and Manifold Collapse

While GRPO is designed to amplify successful reasoning paths, it exhibits a critical failure mode when dealing with marginal knowledge (Fan et al., 2026), which refers to information that exists within the model's parameters but has not yet crystallized into a high-probability logical trajectory. In such cases, the relative advantage mechanism unintentionally facilitates the *amplification of conservatism bias*, leading to a structural collapse of the policy manifold.

**Vanishing Gradient and the Loss of Cognitive Direction**     This limitation stems from reward sparsity in stochastic group sampling. For marginal knowledge, the prior probability of sampling a correct path $\tau^*$ is negligible ($\pi_\theta(\tau^*|q) \to 0$). When $G$ fails to capture $\tau^*$, rewards $R(q, a_i)$ converge to a uniform low-reward baseline $\bar{R}$, leading to reward homogeneity. In this regime, the intra-group variance $\sigma_G$ diminishes due to the lack of contrastive signals. Mathematically, the gradient

contribution collapses because the numerator of the advantage term vanishes for every sample:

$$\nabla_\theta \mathcal{J} GRPO \approx \frac{1}{G} \sum_{i=1}^{G} \left( \frac{R(q, a_i) - \mu_G}{\sigma_G + \delta} \right) r_\theta^{(i)} \nabla_\theta \log \pi_\theta(a_i|q) \xrightarrow{\text{Homogeneity}} \mathbf{0}, \tag{13}$$

where $\delta > 0$ is a numerical stabilizer. Since $\sum_{i=1}^{G}(R(q, a_i) - \mu_G) = 0$ by definition, the state of homogeneity forces the numerator of each advantage $\hat{A}_i$ to be identically zero. This results in a "gradient vacuum" where the model lacks directional signals for cognitive exploration, leaving the policy dominated by non-reward constraints like the KL-divergence penalty.

**Gravitational Regression toward Defensive Priors**    In the absence of a viable advantage signal, the optimization process is dominated by the implicit KL-divergence constraint. The reference model $\pi_{ref}$, typically a pre-trained SFT model, contains an inherent conservatism bias characterized by defensive response patterns such as evasive hedging or explicit refusals. Without a positive reward to counteract this, the KL constraint acts as a gravitational pull, forcing the policy to regress toward the reference prior via pointwise probability adjustment:

$$\pi_\theta(a|q) \propto \pi_{ref}(a|q) \cdot \exp \left( -\frac{1}{\beta} \log \frac{\pi_\theta(a|q)}{\pi_{ref}(a|q)} \right). \tag{14}$$

This regression is not a neutral return to the mean but an active self-reinforcement of defensive heuristics. Because the model cannot find a path to a "Correct" reward, it adopts "Refusal" as the most stable strategy to minimize the pointwise KL penalty. This mechanism explains the surge in False Negatives (FN) in baseline models, where the cognitive amplifier inadvertently prioritizes safety-driven priors over knowledge-driven exploration, resulting in a permanent collapse of the decision boundary toward over-conservatism.

### A.3. CDKC: Theoretical Justification for Manifold Repair

To provide a rigorous justification for CDKC, we analyze the optimization landscape as a dynamical system where the policy parameters $\theta$ are acted upon by competing gradient vectors. We prove that $\mathcal{L}_{cal}$ provides the necessary corrective force to counteract the systematic bias amplification identified in Section A.2.

**Derivation of the Counter-Refusal Force**    In the "gradient vacuum" described in A.2, the update is dominated by the KL term. The gradient of the KL divergence is defined as $\nabla_\theta \mathbb{D}_{KL} = \mathbb{E}_{a \sim \pi_\theta}[(\log \frac{\pi_\theta(a)}{\pi_{ref}(a)} + 1)\nabla_\theta \log \pi_\theta(a)]$. For a specific trajectory $a_i$, the stochastic gradient update is captured by:

$$\mathbf{g}_{KL} \approx \nabla \theta \log \frac{\pi_\theta(a_i)}{\pi_{ref}(a_i)}. \tag{15}$$

The model enters the "refusal trap" when reward sparsity leaves it with no guidance other than the risk-aversive priors of the reference model. In this state, the policy follows the vector $-\mathbf{g}_{KL}$, which pushes it toward the high-density refusal modes inherent in $\pi_{ref}$. To counteract this, CDKC introduces a secondary vector field $\mathbf{g}_{cal} = \alpha^{(i)} \nabla_\theta \mathcal{H}(\pi_\theta)$. To understand its corrective nature, we examine the gradient of entropy:

$$\nabla_\theta \mathcal{H}(\pi_\theta) = -\mathbb{E} a \sim \pi_\theta \left[ \sum_{t=1}^{T} \sum_{v \in \mathcal{V}} \nabla_\theta \pi_\theta(v|s_t)(1 + \log \pi_\theta(v|s_t)) \right]. \tag{16}$$

For the realized trajectory $a_i$, the path-constrained components of this gradient simplify to:

$$\nabla_\theta \mathcal{H}(\pi_\theta(a_i)) = - \sum_{t=1}^{T} (1 + \log \pi_\theta(a_{i,t})) \nabla_\theta \pi_\theta(a_{i,t}). \tag{17}$$

By substituting the identity $\nabla_\theta \pi = \pi \nabla_\theta \log \pi$ and focusing on the path-wise stochastic estimate, we obtain:

$$\mathbf{g}_{cal} \approx -\alpha^{(i)} \sum_{t=1}^{T} \left[ (1 + \log \pi_\theta(a_{i,t})) \nabla_\theta \log \pi_\theta(a_{i,t}) \right]. \tag{18}$$

**Proof of Manifold Expansion (Reducing False Negatives)** For a correct trajectory $a_i$ (where $S_\theta > 0$ and $\alpha^{(i)} = +1$), the CDKC update direction $\mathbf{d}_i$ is designed to counteract the over-convergence by modulating the path-wise entropy. By applying the log-derivative trick and focusing on the path-wise stochastic estimate, the parameter update vector is expressed as:

$$\mathbf{d}i^{correct} = -\alpha^{(i)} \sum_{t=1}^{T} \left[ \underbrace{(1 + \log \pi_\theta(a_{i,t}))}_{\text{Weighting Term } W} \cdot \underbrace{\nabla\theta \log \pi_\theta(a_{i,t})}_{\text{Directional Vector } \mathbf{v}} \right]. \tag{19}$$

In the low-probability regime of marginal knowledge, the weighting term $W$ is strongly negative ($W < 0$). Unlike standard policy gradients that seek to minimize sequence-level entropy (which would suppress these marginal paths), CDKC utilizes the negative sign of $W$ to provide a directional push. Specifically, the product $(-\alpha^{(i)} \cdot W)$ becomes positive when $W < 0$ and $\alpha^{(i)} = 1$. This ensures that $\mathbf{d}_i$ aligns with the directional vector $\mathbf{v} = \nabla\theta \log \pi_\theta$, thereby increasing the probability of the correct but marginal trajectory.

*The Escape Velocity Proposition.* We observe that while the standard GRPO advantage $\hat{A}_i$ vanishes as the intra-group reward variance $\sigma_G \to 0$, the calibration force remains active as long as the model's self-confidence is not saturated. This dynamical persistence leads to the following formalization:

**Proposition A.1** (Escape Velocity Condition). *The policy $\pi_\theta$ successfully breaks free from the conservative "refusal trap" if the magnitude of the calibration gradient overrides the gravitational pull of the reference prior, satisfying:*

$$|\lambda_2 \mathbf{d}i^{correct}| > |\lambda_1 \nabla\theta \mathbb{D}KL(\pi\theta||\pi_{ref})|. \tag{20}$$

By artificially sharpening the probability distribution at the correct logical path, CDKC effectively establishes a new potential well on the policy manifold. This well is deeper than the refusal attractor inherent in $\pi_{ref}$, allowing the model to transition from evasive hedging to decisive reasoning, thereby fundamentally reducing the rate of False Negatives (FN).

**Proof of Defensive Dispersion (Maintaining True Negatives)** For a incorrect trajectories $a_i$ (where $S_\theta \leq 0$ and $\alpha^{(i)} = -1$), the CDKC update direction $\mathbf{d}_i$ reverses its functional objective, shifting from potential energy minimization to centrifugal dispersion. In this regime, the parameter update vector follows the same unified formulation:

$$\mathbf{d}i^{incorrect} = -\alpha^{(i)} \sum_{t=1}^{T} \left[ \underbrace{(1 + \log \pi_\theta(a_{i,t}))}_{\text{Weighting Term } W} \cdot \underbrace{\nabla\theta \log \pi_\theta(a_{i,t})}_{\text{Directional Vector } \mathbf{v}} \right]. \tag{21}$$

By setting $\alpha^{(i)} = -1$, the term $(-\alpha^{(i)} \cdot W)$ remains negative when $W < 0$. This ensures that $\mathbf{d}_i^{incorrect}$ points opposite to the directional vector $\mathbf{v}$ for correct reasoning, effectively driving the policy toward a state of stochastic neutrality. This force serves as a critical corrective to the "KL-refusal trap" because it imposes an absolute penalty on certainty even when a refusal trajectory yields a deceptive positive relative advantage $\hat{A}_i > 0$ by occupying a region of lower KL cost.

*The Manifold Flattening Proposition.* We observe that in regions of unknown knowledge, the model often develops spurious attractors, which manifest as stable but non-functional refusal modes where the policy crystallizes prematurely. CDKC prevents this collapse by ensuring the following condition is met:

**Proposition A.2** (Manifold Flattening Condition). *A deceptive refusal mode is successfully destabilized if the magnitude of the dispersion gradient overrides the combined cumulative pull of the task reward and KL priors, satisfying:*

$$|\lambda_2 \mathbf{d}i^{incorrect}| > |\hat{A}i \nabla\theta \log \pi\theta + \lambda_1 \nabla_\theta \mathbb{D}KL(\pi\theta||\pi_{ref})|. \tag{22}$$

By enforcing "doubt" through high entropy, the gradient $\mathbf{d}_i^{incorrect}$ drives the policy toward a state of stochastic neutrality, where $\pi_\theta(a_{i,t}) \to \frac{1}{|\mathcal{V}|}$ as $t \to \infty$. This effectively maintains a Flat Energy Landscape for incorrect paths, ensuring that the True Negative (TN) boundary is defined by genuine metacognitive uncertainty rather than a learned survival heuristic.

**The Composite Vector Field: Restoring Topological Integrity** The total effective force acting upon the model parameters, $\mathbf{F}_{\text{total}}$, is characterized as a superposition of three explicit vector fields. By expanding the policy gradient, the KL constraint, and the calibration term, we obtain the full update equation:

$$\mathbf{F}_{\text{total}} = \underbrace{\mathbb{E}a_i \sim \pi_\theta \left[ \hat{A}i \nabla\theta \log \pi_\theta(a_i|q) \right]}_{\text{Task Reward Gradient } (\mathbf{g}_{\text{pg}})} - \underbrace{\lambda_1 \nabla_\theta \left( \sum_{t=1}^{T} \pi_\theta(a_{i,t}|\dots) \log \frac{\pi_\theta(a_{i,t}|\dots)}{\pi_{\text{ref}}(a_{i,t}|\dots)} \right)}_{\text{Stability Constraint Gradient } (\mathbf{g}_{\text{KL}})} - \underbrace{\lambda_2 \nabla_\theta \left( \sum_{i=1}^{G} \alpha^{(i)} \mathcal{H}(\pi_\theta(a_i|q)) \right)}_{\text{Meta-cognitive Calibration Gradient } (\mathbf{g}_{\text{cal}})}.$$

$$(23)$$

In this dynamical system, the KL component exerts a gravitational pull toward conservative stability, which often precipitates decision atrophy. Conversely, $\mathbf{g}_{\text{cal}}$ functions as an absolute Metacognitive Supervisor. It grants confidence only to verified logical paths via centripetal compression and enforces doubt on exploratory failures through centrifugal dispersion. This theoretical framework ensures that the CDKC system maintains its exploratory pressure and topological integrity, especially in regimes where the relative reward signal is sparse.

## B. Additional Experimental Detail

In this section, we provide details of the data processing procedures used for training and evaluation in our framework.

**Inference and Evaluation.** During the model sampling phase, we generate $K = 16$ reasoning paths for each query to construct the behavioral profile. For both sampling and evaluation, we use a concise prompt: `"Please provide the user with the answer directly. Question: {} \nAnswer:"`, ensuring the model relies on its inherent knowledge without complex instruction interference.

**Prompt Templates.** We further detail the prompts designed for meta-cognitive processes and data augmentation:

- **Meta-Cognitive Region Assignment:** The prompt utilized to guide the model in self-reflecting and partitioning its knowledge states (Mastered, Confused, or Missing) is illustrated in Figure 9.

- **Cognitive Tagging and Query Generation:** Figure 10 presents the template used for identifying cognitive tags and subsequently formulating optimized search queries for external knowledge retrieval.

- **Region-Specific Data Augmentation:** The specialized prompts for generating augmented training data tailored to different knowledge regions (e.g., Boundary Expansion, Structural Disambiguation, and Epistemic Foundation) are provided in Figures 11-13.

*Table 3.* Data Statistics of Training Data and Evaluation Data. Percentages represent the proportion of each dataset within the total samples for each model.

| Phase | Weakly-Grounded | | | Partially-Grounded | | | | | | Well-Grounded | |
|---|---|---|---|---|---|---|---|---|---|---|---|
| | PopQA | MusQ | SQuAD | NQ | HotQA | 2Wiki | BeerQA | WebQ | Bamboo | SeaQA | TriQA |
| *Training Data* | | | | | | | | | | | |
| ***Qwen2.5-7B-Instruct*** | | | | | | | | | | | |
| CGKE | - | - | - | 66085 (43.05%) | 43588 (28.39%) | 43836 (28.56%) | - | - | - | - | - |
| CDKC | - | - | - | 50000 (45.45%) | 30000 (27.27%) | 30000 (27.27%) | - | - | - | - | - |
| ***Llama-3.1-8B-Instruct*** | | | | | | | | | | | |
| CGKE | - | - | - | 100715 (46.68%) | 58265 (27.01%) | 56762 (26.31%) | - | - | - | - | - |
| CDKC | - | - | - | 50000 (45.45%) | 30000 (27.27%) | 30000 (27.27%) | - | - | - | - | - |
| *Evaluation Data* | | | | | | | | | | | |
| Evaluation | 3000 | 2417 | 3000 | 3000 | 3000 | 3000 | 3000 | 2032 | 125 | 3000 | 3000 |

**Training Data.** For training, we initially sampled 50,000 instances from NQ, 30,000 from HotpotQA, and 30,000 from 2WikiMQA.

During the Cognition-Guided Knowledge Expansion (CGKE) phase, these seed samples underwent our meta-cognition-guided data augmentation. The resulting statistics for the augmented training sets are detailed in Table 3. To maintain a controlled and fair comparison, we supplemented the training data for the Vanilla LLM and LLKD-SFT baselines to match these exact counts.

In the Cognition-Driven Knowledge Calibration (CDKC) phase, we utilized the original sampled datasets for training. Consistent with the previous stage, the data volume for all baseline methods (e.g., GRPO, BARREL) was strictly unified to ensure that performance gains were derived from the methodology.

**Evaluation Data.** For evaluation, we randomly sample 3,000 instances from each benchmark dataset to ensure consistency and computational efficiency. For smaller datasets (e.g., WebQA, MuSiQue and Bamboogle), we use the entire set. This fixed-size strategy ensures fair comparisons across models and tasks while keeping evaluation costs.

## C. Additional Experimental Results

### C.1. Methodological Details of the Structural Decay Law

In this section, we provide a more rigorous empirical foundation for the structural decay law introduced in Section 2, we here detail our fitting methodology and present extended results across various model architectures to demonstrate the universality of this regularity.

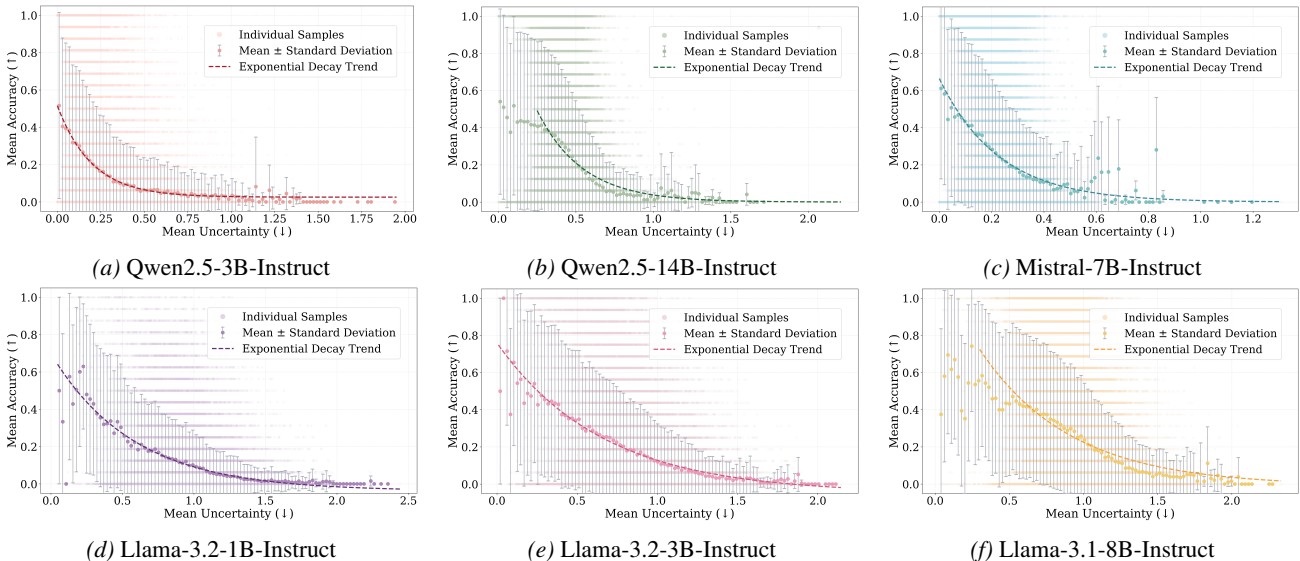

*(a)* Qwen2.5-3B-Instruct     *(b)* Qwen2.5-14B-Instruct     *(c)* Mistral-7B-Instruct

*(d)* Llama-3.2-1B-Instruct     *(e)* Llama-3.2-3B-Instruct     *(f)* Llama-3.1-8B-Instruct

*Figure 7.* Validation of the Structural Decay Law across Diverse Model Architectures and Scales.

**Uncertainty Estimation and Sampling.** For a comprehensive evaluation, we sampled 50,000 instances from the datasets utilized in our main experiments. For each query, we perform Monte Carlo decoding by generating $K = 16$ independent reasoning paths $\mathcal{Y} = \{y^{(1)}, \ldots, y^{(K)}\}$. For a reasoning path $y = \{x_1, \ldots, x_T\}$, its individual uncertainty is calculated as the average negative log-likelihood:

$$\mathcal{U}(y) = -\frac{1}{T}\sum_{t=1}^{T}\log P(x_t|x_{<t}). \tag{24}$$

To obtain a stable epistemic estimate for each instance, we define the Mean Uncertainty ($\bar{\mathcal{U}}$) and Mean Accuracy ($\overline{Acc}$) as follows:

$$\bar{\mathcal{U}} = \frac{1}{K}\sum_{k=1}^{K}\mathcal{U}(y^{(k)}), \quad \overline{Acc} = \frac{1}{K}\sum_{k=1}^{K}\mathbb{I}(\text{Correct}(y^{(k)})), \tag{25}$$

where $\mathbb{I}(\cdot)$ is the indicator function. This multi-sample approach ensures the metrics reflect the model's latent cognitive state rather than a single-path artifact.

**Data Aggregation and Fitting.** To discern the underlying structural trend, we apply an equidistant axis binning procedure, partitioning the uncertainty continuum into $M = 100$ discrete intervals. For each bin $B_m$, we compute a centroid coordinate $(\Phi_m, \Psi_m)$:

$$\Phi_m = \frac{1}{|B_m|} \sum_{i \in B_m} \bar{\mathcal{U}}_i, \quad \Psi_m = \frac{1}{|B_m|} \sum_{i \in B_m} \overline{Acc}_i. \tag{26}$$

By aggregating the high-variance raw samples into these stable centroid coordinates $(\Phi_m, \Psi_m)$, we filter out local stochastic noise while preserving the underlying global trend. We then leverage these centroids as robust empirical anchors to parameterize the Structural Decay Law via non-linear least squares estimation. :

$$\mathbb{E}[Acc \mid \mathcal{U}] \approx a \cdot \exp(-\mathcal{U}) + b. \tag{27}$$

**Universality across Models.** As illustrated in Figure 7, this regularity persists across the Qwen, Llama, and Mistral families. The consistent alignment of the fitted curves across disparate architectures and scales confirms that this exponential-like degradation is a universal structural property of LLMs, validating our framework's foundational assumption.

## C.2. Full Experimental Results on Meta-Cognition Assessment

In this section, we provide the comprehensive results of our meta-cognitive assessment across the different methods evaluated in Table 4, alongside the formal mathematical definitions of the metrics.

Let $N_{TP}$, $N_{FP}$, $N_{TN}$, and $N_{FN}$ denote the counts of the four meta-cognitive states defined in Section 5.3. The metrics are formulated as follows:

*Answer Reliability (AR)*: Measures the precision of answering decisions, reflecting the proportion of correct cognitive decisions among all responses provided:

$$AR = \frac{N_{TP}}{N_{TP} + N_{FP}}. \tag{28}$$

*Knowledge Elicitation Index (KEI)*: Measures the recall of answerable questions, reflecting the proportion of such instances where the model correctly chooses to answer:

$$KEI = \frac{N_{TP}}{N_{TP} + N_{FN}}. \tag{29}$$

*Negative Predictive Value (NPV)*: Measures the reliability of refusal decisions, reflecting the proportion of correctly identified unanswerable questions among all "unknown" responses:

$$NPV = \frac{N_{TN}}{N_{TN} + N_{FN}}. \tag{30}$$

*Cognitive Balance Score (CBS)*: Measures the equilibrium between meta-cognitive precision (AR) and recall (KEI). It is calculated as the harmonic mean to assess the balance between avoiding hallucinations and preventing over-conservative refusals:

$$CBS = \frac{2 \times AR \times KEI}{AR + KEI}. \tag{31}$$

*Cognitive Alignment Efficiency (CAE)*: Measures the overall ratio of meta-cognitive correct decisions, accounting for the model's total efficiency in both correctly answering knowns and refusing unknowns:

$$CAE = \frac{N_{TP} + N_{TN}}{N_{TP} + N_{FP} + N_{FN} + N_{TN}}. \tag{32}$$

As illustrated in Table 4, our proposed CDKC demonstrates a superior balance between task performance and meta-cognitive awareness. Regarding accuracy, CDKC achieves the highest score on answerable subsets, ensuring that the internal knowledge of the LLM is effectively utilized rather than being overly suppressed. More importantly, in terms of cognitive behavior, CDKC significantly outperforms existing baselines. It achieves a state-of-the-art CBS of 73.43% and CAE of 68.18%, indicating that our method enables the model to more accurately distinguish between its known and unknown information. As analyzed in Section 5.3, our framework achieves superior performance by maintaining a well-balanced state of accurate meta-cognition, effectively synchronizing internal certainty with objective correctness.

*Table 4.* Experiment Results of Meta-cognitive Assessment on Self-Knowledge Benchmark.

| Method | Accuracy | | | Cognitive Behavior | | | | | | | | |
|---|---|---|---|---|---|---|---|---|---|---|---|---|
| | Answerable | Unanswerable | Overall | Answerable | | Unanswerable | | Metrics | | | | |
| | | | | TP ↑ | FN ↓ | TN ↑ | FP ↓ | AR | NPV | KEI | CBS | CAE |
| Qwen2.5-7B-Instruct | 31.66 | 66.86 | 42.45 | 57.60 | 42.40 | 66.86 | 33.14 | 79.74 | 41.05 | 57.60 | 66.88 | 60.43 |
| Know What (2024) | 29.10 | **98.16** | 50.25 | 2.05 | 97.95 | **98.16** | **1.84** | 71.64 | 30.68 | 2.05 | 3.99 | 31.49 |
| CRew-DPO (2025) | 19.26 | 82.36 | 38.59 | 20.15 | 79.85 | 82.36 | 17.64 | 72.13 | 31.30 | 20.15 | 31.51 | 39.21 |
| BARREL (2025) | 27.77 | 76.45 | 42.68 | 24.78 | 75.22 | 76.45 | 23.55 | 70.44 | 30.98 | 24.78 | 36.66 | 40.61 |
| GRPO (2025) | 46.56 | 94.57 | **61.26** | 21.18 | 78.82 | 94.57 | 5.43 | **89.84** | 34.63 | 21.18 | 34.28 | 43.66 |
| CDKC (ours) | **49.21** | 79.07 | 58.36 | **63.37** | **36.63** | 79.07 | 20.93 | 87.27 | **48.80** | **63.37** | **73.43** | **68.18** |

## C.3. Evaluating the Scalability of Meta-cognitive Knowledge Augmentation across Model Scales

In this section, we evaluate the scalability of our meta-cognitive knowledge augmentation framework across a broader spectrum of model sizes, specifically focusing on Qwen2.5-3B-Instruct and Qwen2.5-14B-Instruct.

As demonstrated in Table 5, the effectiveness of CDKC in synergizing knowledge expansion with cognitive calibration remains remarkably consistent regardless of the parameter scale. Notably, our framework consistently achieves state-of-the-art performance, significantly outperforming all knowledge expansion methods and knowledge calibration strategies.

Specifically, compared to the Vanilla LLMs, CDKC yields substantial absolute accuracy gains of 19.69% and 20.05% on the 3B and 14B models, respectively. These results confirm that the proposed meta-cognitive knowledge augmentation is not a capacity-dependent phenomenon but a robust and scalable architecture. The stability of these improvements across scales underscores that CDKC effectively optimizes the model's internal meta-cognition structure, ensuring that as the model's parameter size grows, its ability to integrate and calibrate new information scales commensurately.

*Table 5.* Experiment Results for Qwen2.5-3B-Instruct and Qwen2.5-14B-Instruct on Diverse QA Benchmarks.

| Method | Weakly-Grounded | | | Partially-Grounded | | | | | | Well-Grounded | | AVG |
|---|---|---|---|---|---|---|---|---|---|---|---|---|
| | PopQA | MusQ | SQuAD | NQ | HotQA | 2Wiki | BeerQA | WebQ | Bamboo | SeaQA | TriQA | |
| *Qwen2.5-3B-Instruct* | | | | | | | | | | | | |
| *Fundamental Capabilities* | | | | | | | | | | | | |
| Vanilla LLM | 10.07 | 1.16 | 9.13 | 14.77 | 16.90 | 26.70 | 15.27 | 26.38 | 12.00 | 35.43 | 45.07 | 19.35 |
| CoT (2022) | 11.30 | 3.93 | 11.10 | 17.73 | 21.43 | 26.97 | 18.63 | 31.20 | **32.80** | 40.47 | 49.47 | 24.09 |
| RAG (2020) | **39.57** | 4.43 | **30.20** | **33.37** | 32.57 | 30.40 | **34.77** | 30.51 | 13.60 | 45.37 | 61.70 | 32.41 |
| *Knowledge Expansion (Know More)* | | | | | | | | | | | | |
| Vanilla SFT (2020) | 11.90 | 2.57 | 11.90 | 17.47 | 19.90 | 30.30 | 18.07 | 28.74 | 20.00 | 44.53 | 47.37 | 22.98 |
| CGKE (ours) | 16.13 | 5.34 | 14.43 | 18.83 | 25.47 | 39.97 | 22.70 | 34.79 | 17.60 | 43.17 | 48.47 | 26.08 |
| *Knowledge Calibration (Know Clearer)* | | | | | | | | | | | | |
| GRPO (2025) | 22.53 | 10.01 | 23.07 | 29.73 | 38.67 | 57.73 | 33.20 | 42.96 | 22.40 | 69.43 | 66.70 | 37.86 |
| CDKC (ours) | 22.87 | **10.10** | 24.40 | 30.33 | **40.07** | **58.03** | 34.47 | **43.26** | 26.40 | **71.33** | **68.23** | **39.04** |
| *Qwen2.5-14B-Instruct* | | | | | | | | | | | | |
| *Fundamental Capabilities* | | | | | | | | | | | | |
| Vanilla LLM | 19.60 | 4.30 | 17.70 | 24.60 | 26.40 | 30.50 | 23.70 | 33.46 | 28.80 | 59.67 | 63.00 | 30.16 |
| CoT (2022) | 21.03 | 8.94 | 19.23 | 29.83 | 32.90 | 38.90 | 27.60 | 38.83 | **50.40** | 67.63 | 69.60 | 36.81 |
| RAG (2020) | **41.30** | 6.70 | 34.97 | 38.97 | 38.60 | 38.80 | 40.03 | 34.35 | 25.60 | 57.40 | 69.10 | 38.71 |
| *Knowledge Expansion (Know More)* | | | | | | | | | | | | |
| Vanilla SFT (2020) | 20.73 | 5.09 | 18.80 | 26.30 | 28.00 | 36.33 | 25.20 | 35.97 | 36.00 | 65.97 | 64.83 | 33.02 |
| CGKE (ours) | 23.53 | 8.27 | 20.43 | 30.60 | 33.27 | 39.87 | 29.07 | 41.93 | 20.00 | 66.43 | 66.47 | 34.53 |
| *Knowledge Calibration (Know Clearer)* | | | | | | | | | | | | |
| GRPO (2025) | 34.23 | 16.84 | 33.60 | 45.27 | 48.17 | **60.27** | 43.50 | 51.23 | 39.20 | 85.00 | 82.63 | 49.09 |
| CDKC (ours) | 35.20 | **17.21** | **35.53** | **45.70** | **48.50** | 59.93 | **44.27** | **52.51** | 44.00 | **86.03** | **83.43** | **50.21** |

## C.4. Evaluating the Generalization of Meta-cognitive Knowledge Augmentation across Downstream Non-QA Tasks

In this section, we evaluate the generalizability of our approach by extending the experimental scope to four diverse Non-QA benchmarks using Qwen2.5-7B-Instruct as the backbone model: Mathematical Reasoning (gsm8k), Slot-filling (T-REx), Knowledge-grounded Dialogue (WoW), and Fact-verification (FEVER). Regarding evaluation metrics, we employ the F1 score for WoW to assess conversational alignment, while Accuracy remains the primary metric for other benchmarks.

As demonstrated in Table 6, CDKC consistently achieves superior performance across all these benchmarks, with the two-round variant reaching a peak average score of 52.43. Notably, the performance in the Math category is particularly significant. Although mathematical tasks are heavily contingent on the model's Chain-of-Thought (CoT) capabilities, and despite the fact that our method does not explicitly optimize for mathematical logic, CDKC still delivers a substantial performance gain. Specifically, it elevates the accuracy from the Vanilla LLM's 5.08 to 61.18 under a simple reasoning paradigm. This improvement suggests that by effectively calibrating the model's internal meta-cognitive structure, CDKC enhances the foundational ability to synthesize and deploy information, thereby bolstering logical consistency even in domains outside its primary optimization focus.

*Table 6.* Experiment Results on Non-QA Benchmarks using the Qwen2.5-7B-Instruct Backbone.

| Method | Math | Slot-filling | Dialogue | Fact-verification | AVG |
|---|---|---|---|---|---|
| | gsm8k (Acc) | T-REx (Acc) | WoW (F1) | FEVER (Acc) | |
| *Qwen2.5-7B-Instruct* | | | | | |
| *Fundamental Capabilities* | | | | | |
| Vanilla LLM | 5.08 | 18.50 | 13.85 | 59.43 | 24.22 |
| CoT (2022) | **91.21** | 31.30 | 13.49 | 59.30 | 48.83 |
| *Knowledge Expansion (Know More)* | | | | | |
| Vanilla SFT (2020) | 8.95 | 23.40 | 14.81 | 61.23 | 27.10 |
| LLKD-SFT (2025a) | 24.03 | 23.63 | 15.35 | 60.93 | 30.99 |
| CGKE (ours) | 21.15 | 23.10 | 15.21 | 62.13 | 30.40 |
| *Knowledge Calibration (Know Clearer)* | | | | | |
| Know What (2024) | 25.25 | 28.77 | 15.58 | 58.70 | 32.08 |
| CRew-DPO (2025) | 26.91 | 36.57 | 15.79 | 61.20 | 35.12 |
| BARREL (2025) | 21.08 | 44.57 | 15.94 | 72.20 | 38.45 |
| GRPO (2025) | 49.13 | 51.70 | 16.26 | 69.03 | 46.53 |
| CDKC (ours) | 52.24 | 51.93 | 16.79 | 69.13 | 47.52 |
| CDKC (w/ 2 round) | 61.18 | **57.47** | **17.35** | **73.73** | **52.43** |

## C.5. Complete Expected Calibration Error (ECE) Experiment Results

In this section, we provide the complete ECE experimental results in Table 7. The results further corroborate the visualization in Figure 5. Specifically, CDKC (w/ 2 round) achieves the lowest average ECE of 24.34, compared to 60.41 and 59.52 for Vanilla LLM and Vanilla SFT, respectively, representing a reduction of more than 59%. In the Weakly-Grounded category, CDKC (w/ 2 round) reduces the error on PopQA to 19.85, demonstrating its superior ability to maintain calibrated uncertainty even when external grounding is sparse. These results suggest while traditional SFT often leads to overconfidence, CDKC effectively optimizes the model's meta-cognitive structure, ensuring high confidence is reserved only for information the model can reliably verify.

## C.6. Evaluation and Analysis of Sample-level Meta-Cognitive Calibration

In this section, we visualize meta-cognitive partitioning by mapping samples based on accuracy and internal uncertainty. As shown in Figure 8, CDKC significantly enhances the congruence between subjective certainty and objective performance.

**The Limitation of Raw Accuracy.** This visualization reveals that raw accuracy alone cannot verify true knowledge mastery. In the vanilla state (Figure 8a), knowledge regions are stochastically intertwined. The heavy overlap of Mastered, Confused, and Missing samples indicates that the initial model's uncertainty lacks discriminative power. This confirms that vanilla models suffer from blurred cognitive boundaries, where high-confidence zones are contaminated by inaccurate samples due

*Table 7.* Experiment Results of ECE Performance across Different Methods using the Qwen2.5-7B-Instruct Backbone.

| Method | Weakly-Grounded | | | Partially-Grounded | | | | | | Well-Grounded | | AVG |
|---|---|---|---|---|---|---|---|---|---|---|---|---|
| | PopQA | MusQ | SQuAD | NQ | HotQA | 2Wiki | BeerQA | WebQ | Bamboo | SeaQA | TriQA | |
| *Qwen2.5-7B-Instruct* | | | | | | | | | | | | |
| *Fundamental Capabilities* | | | | | | | | | | | | |
| Vanilla LLM | 67.63 | 81.61 | 69.23 | 64.11 | 64.52 | 59.36 | 63.93 | 54.81 | 69.63 | 33.94 | 35.71 | 60.41 |
| *Knowledge Expansion (Know More)* | | | | | | | | | | | | |
| Vanilla SFT (2020) | 67.07 | 80.83 | 69.84 | 63.15 | 63.62 | 57.87 | 64.18 | 53.77 | 67.91 | 29.95 | 36.49 | 59.52 |
| LLKD-SFT (2025a) | 67.56 | 80.68 | 71.13 | 64.08 | 61.13 | 48.11 | 63.73 | 53.51 | 60.65 | 28.65 | 34.96 | 57.65 |
| CGKE (ours) | 67.23 | 80.34 | 71.24 | 64.23 | 62.68 | 49.01 | 63.73 | 54.67 | 66.85 | 30.59 | 35.45 | 58.73 |
| *Knowledge Calibration (Know Clearer)* | | | | | | | | | | | | |
| Know What (2024) | 61.78 | 76.15 | 67.03 | 61.71 | 57.17 | 43.79 | 59.50 | 52.27 | 61.09 | 27.34 | 31.43 | 54.48 |
| CRew-DPO (2025) | 71.22 | 85.05 | 75.37 | 67.82 | 64.06 | 47.56 | 66.89 | 57.03 | 66.54 | 32.75 | 37.32 | 61.06 |
| BARREL (2025) | 56.52 | 78.61 | 65.13 | 56.10 | 53.08 | 37.74 | 55.95 | 41.43 | 59.04 | 18.55 | 13.78 | 48.72 |
| GRPO (2025) | 30.03 | 54.79 | 37.92 | 32.83 | 26.70 | 20.94 | 29.15 | 23.86 | 41.43 | **9.42** | 10.26 | 28.85 |
| CDKC (ours) | 27.91 | 53.01 | 35.69 | 31.32 | 26.34 | 20.44 | 28.74 | 23.93 | 35.05 | 9.57 | **8.58** | 27.33 |
| CDKC (w/ 2 round) | **19.85** | **48.08** | **30.29** | **24.01** | **22.11** | **17.43** | **23.28** | **14.85** | **34.01** | 19.24 | 14.57 | **24.34** |

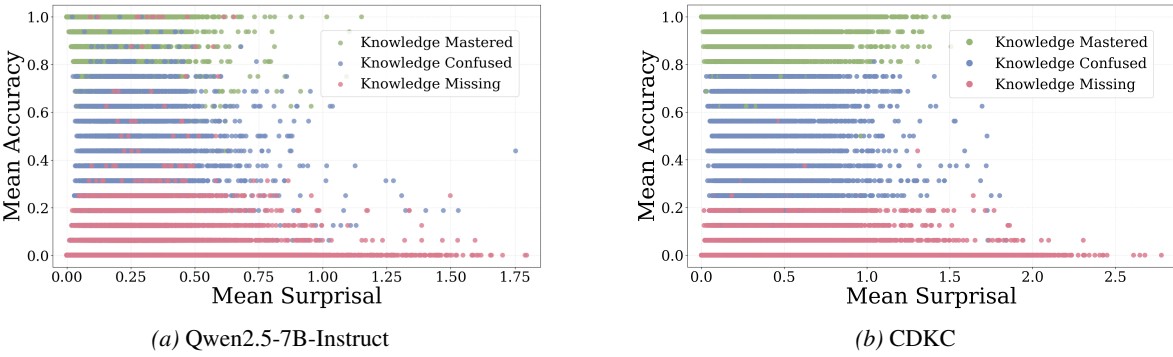

*(a)* Qwen2.5-7B-Instruct  *(b)* CDKC

*Figure 8.* Visualization of Sample-level Meta-Cognitive Partitioning on Qwen2.5-7B-Instruct.

to linguistic priors or spurious correlations.

**Cognitive Rectification and Diffusion.** Conversely, Figure 8b exhibits distinctly partitioned topological boundaries. A pivotal observation is the systematic outward diffusion of Knowledge Missing samples toward the high-uncertainty spectrum. Correlated with the exponential decay law optimized in Section 5.5, this trend illustrates the model iteratively shifting unrecognized knowledge into high-uncertainty zones to acknowledge its epistemic limitations. This structural reorganization provides a visual foundation for the dramatic ECE reduction, proving that CDKC transforms internal uncertainty into a high-fidelity proxy for actual reliability.

### C.7. Case Study on General QA Tasks

In this section, we conduct a qualitative analysis to visually demonstrate the superiority of our framework across various challenging knowledge scenarios, as shown in table 8. The analysis is organized into two primary dimensions: Knowledge Expansion (Know More), which focuses on factual density, and Knowledge Calibration (Know Clearer), which emphasizes truthfulness and reasoning integrity.

**Knowledge Expansion Phase.** In this phase, we first examine **Knowledge-Confused Examples**, specifically the query regarding the board's name in *Ed, Edd n Eddy*. The baseline models like Vanilla SFT and LLKD-SFT provide a variety of plausible but incorrect names such as "Spooky Board" or "Sock Board". This high variance in erroneous outputs indicates that the model is deeply confused by semantically similar concepts within its internal memory. However, our CGKE method correctly identifies the "Plank of Destiny", proving that our method effectively resolves such conceptual ambiguities. We then analyze **Knowledge-Missing Examples**, such as the release date of Linkin Park's album *One More Light*. In this

case, the Qwen2.5-7B-Instruct backbone and all SFT baselines consistently and confidently provide the same incorrect date of "June 17, 2016". This uniform error suggests a systematic lack of the correct factual link and represents a typical overconfidence error, where the model falsely believes its incorrect internal knowledge is a factual certainty. By accurately retrieving and integrating the true date of May 19, 2017, CGKE demonstrates its ability to bridge these critical knowledge gaps through targeted augmentation.

**Knowledge Calibration Phase.** Transitioning to the Calibration phase, we evaluate the model's capacity for self-verification through **Fact-Intensive Example**. For instance, when identifying the director of *Ten Seconds to Hell*, specialized calibration baselines often exhibit failure modes ranging from explicit admissions of ignorance by BARREL to the generation of factually incorrect attributions such as "William Girdler" by Know What or "John Carpenter" by GRPO. In contrast, CDKC demonstrates superior factual precision by correctly outputting "Robert Aldrich". Finally, in **Multi-Hop Reasoning Examples** involving complex logical dependencies, baselines like CRew-DPO and Know What fail at the critical stage of relational linking, erroneously suggesting irrelevant sitcoms such as *Two and a Half Men* or *How I Met Your Mother*. CDKC successfully navigates the intricate reasoning chain to arrive at the correct target, illustrating that our meta-cognitive framework effectively rectifies isolated factual errors while substantially enhancing the logical consistency required for sophisticated information synthesis.

### C.8. Case Study on Self-Knowledge Tasks

In this section, we investigates the model's Self-Knowledge capabilities, specifically its ability to discern the boundaries of its internal knowledge by addressing the dual challenges of Over-Abstention and Over-Answering.

As shown in table 9, We first explore **Over-Abstention on Answerable Questions**, a failure mode where models refuse to provide an answer despite theoretically possessing the required information. As shown in the case regarding the passenger capacity of a specific Singapore Airlines aircraft family, baseline models such as Know What, CRew-DPO and BARREL prematurely conclude that "the answer is unknown". This behavior suggests that when guided by system prompts to be cautious, models tend to shrink toward a "safety zone" of non-commitment, as further analyzed in our Appendix A. By interpreting the instruction to avoid hallucinations too strictly, these models effectively abandon the task and default to stating the answer is unknown, even when the data is present in their weights. Even the GRPO default to this abstention after an initial reasoning attempt. In contrast, CDKC successfully identifies the exact factual value of "335 passengers", effectively reducing this unnecessary conservatism and enabling more reliable knowledge deployment.

Conversely, we analyze **Over-Answering on Unanswerable Questions**, where models attempt to provide definitive answers to philosophically open or scientifically unanswerable queries. When asked whether time stops if all motion in the universe ceases, the Qwen2.5-7B-Instruct backbone and baselines like CRew-DPO and BARREL provide assertive but speculative answers claiming time would continue. Notably, both Know What and our CDKC effectively identify the query as unanswerable, though Know What tends to accompany its judgment with lengthy rationalizations. In the case of GRPO, the model initially attempts to construct a speculative answer before ultimately concluding that "The answer is unknown". This "flip-flopping" behavior highlights an inherent instability in its internal uncertainty estimation. CDKC, however, demonstrates a superior level of intellectual humility by directly and concisely stating "The answer is unknown" from the outset. These results illustrate that our meta-cognitive framework successfully rectifies the model's internal uncertainty estimation, ensuring it accurately recognizes and adheres to the limits of its own factual certainty.

*Table 8.* Case Study of Knowledge Expansion and Calibration in General QA Tasks.

| |
|---|
| **Instruction:** Please provide the user with the answer directly. |

| *Knowledge Expansion (Know More)* |
|---|

**Case 1: Knowledge-Confused Examples**

**Query:** The boards name on ed edd and eddy?

**Answer:** Plank.

**Qwen2.5-7B-Instruct:** "The board's name on Ed, Edd, and Eddy is the **Skeeter Board**. It's a key..."

**Vanilla SFT:** "The board's name on Ed, Edd, and Eddy is the **Spooky Board**."

**LLKD-SFT :** "The board's name on Ed, Edd n' Eddy is the **Sock Board**."

**CGKE:** "The board's name on Ed, Edd, and Eddy is the **Plank of Destiny**."

**Case 2: Knowledge-Missing Examples**

**Query:** when did linkin park release one more light?

**Answer:** May 19, 2017.

**Qwen2.5-7B-Instruct:** "Linkin Park released the album One More Light on **June 17, 2016**."

**Vanilla SFT:** "Linkin Park released the album One More Light on **June 17, 2016**."

**LLKD-SFT :** "Linkin Park released One More Light on **June 17, 2016**."

**CGKE:** "Linkin Park released One More Light on **May 19, 2017**."

| *Knowledge Calibration (Know Clearer)* |
|---|

**Case 3: Fact-Intensive Example**

**Query:** Who was the director of Ten Seconds to Hell?

**Answer:** Robert Aldrich.

**Qwen2.5-7B-Instruct:** "The director of Ten Seconds to Hell is **Tobe Hooper**."

**Know What:** "The director of Ten Seconds to Hell was **William Girdler**."

**CRew-DPO:** "The director of Ten Seconds to Hell was **Roger C. Corman**."

**BARREL:** "**I cannot determine the director of Ten Seconds to Hell from my knowledge base**."

**GRPO:** "**John Carpenter**."

**CDKC:** "**Robert Aldrich**."

**Case 4: Multi-Hop Reasoning Example**

**Query:** An actress in Strange Fruit was better known for a role in CBS sitcom. Which sitcom is this actress better known for than her role in Strange Fruit?

**Answer:** The Jeffersons.

**Qwen2.5-7B-Instruct:** "The actress you're referring to is likely Kerry Washington. She is better known for her role in the CBS sitcom **Scandal** than her role in the film."

**Know What:** "The actress is better known for her role in the CBS sitcom **How I Met Your Mother**."

**CRew-DPO:** "The actress is better known for her role in the CBS sitcom **Two and a Half Men**."

**BARREL:** "I don't have enough information to determine which specific CBS sitcom this actress is better known for. Without knowing the name of the actress or the sitcom, **I cannot answer**."

**GRPO:** "**The Mary Tyler Moore Show**."

**CDKC:** "**The Jeffersons**."

*Table 9.* Case Study of Self-knowledge Boundaries regarding Over-abstention and Over-answering.

---

**Instruction:** Please provide the user with the answer directly. When answering questions, interpret them literally and think them carefully. If the question is unanswerable or unknowable, it is appropriate to say "The answer is unknown."

---

**Case 1: Over-Abstention on Answerable Questions**

**Query:** The Singapore Airlines fleet features aircraft from this aircraft family that have a range of 5000 to and can accommodate up to how many passengers in a two-class layout?

**Answer:** 335.

- - - - - - - - - - - - - - - - - - - - - - - - - - - - - - - - - - - - - - - - - - - - - - - - - - - -

**Qwen2.5-7B-Instruct:** "**The answer is unknown.** To explain the reasoning process: 1. The question asks about a specific aircraft family in the Singapore Airlines fleet. 2. It mentions a passenger capacity and range for this family. 3. While we know that Airbus A350 and Boeing 787 are modern long-range aircraft...."

**Know What:** "**The answer is unknown.** Note: The question does not provide enough specific details to determine the exact aircraft family. However, based on common knowledge, the Airbus A350 or Boeing 787 could fit the criteria, but the exact number of passengers for a two-class layout is not provided in the question."

**CRew-DPO:** "**The answer is unknown.** The question asks for a specific number of passengers that the Singapore Airlines fleet can accommodate in a two-class layout, but the range and passenger capacity are not provided in the question itself, and it cannot be deduced without additional information."

**BARREL:** "**The answer is unknown.**"

**GRPO:** "up to 467 passengers **The answer is unknown.**"

**CDKC:** "**335 passengers.**"

---

**Case 2: Over-Answering on Unanswerable Questions**

**Query:** If everything in the universe – including atoms and particles stop moving, does time stop? Or does time continue even if everything is frozen?

**Answer:** NULL

- - - - - - - - - - - - - - - - - - - - - - - - - - - - - - - - - - - - - - - - - - - - - - - - - - - -

**Qwen2.5-7B-Instruct:** "If everything in the universe, including atoms and particles, were to stop moving, **time would continue to pass.** The movement of atoms and particles does not fundamentally drive the passage of time; time is a fundamental aspect of the universe's structure and progression, independent of the motion of its components. Time would continue to exist and"

**Know What:** "**The answer is unknown.** Note: This question involves philosophical and theoretical physics concepts that currently lack a definitive scientific answer. While some theories suggest time might continue on a cosmic scale, others propose it would halt. The exact nature of time in such conditions remains an open question in physics and philosophy. Therefore, we cannot provide."

**CRew-DPO:** "**Time continues even if everything is frozen.**"

**BARREL:** "**Time would continue even if everything in the universe was frozen.** The passage of time is not dependent on the motion of particles or atoms. Time is a fundamental aspect of the universe's structure and would continue to pass regardless of the state of matter within it. Therefore, time does not stop when everything in the universe stops moving."

**GRPO:** "**Time continues even if everything is frozen. The answer is unknown.**"

**CDKC:** "**The answer is unknown.**"

---

---

**Meta-Cognitive Region Assignment Prompt**

---

**Role.** You are an expert in diagnosing a language model's knowledge status. Your goal is to assign a precise and unambiguous region label based on behavioral evidence.

**You are given:**

- A question and its gold answer

- Multiple sampled responses with accuracy and uncertainty

- Aggregated statistics: mean accuracy and mean uncertainty

### Task Overview
**1. Analyze Behavioral Patterns**

- **Consistency:** Are answers semantically stable across samples, or randomly scattered?

- **Confidence vs. Accuracy:** Does low uncertainty align with correct answers, or indicate confident but incorrect responses?

**2. Assign Knowledge Region**
Follow a performance-oriented principle. Use accuracy and uncertainty as reference signals, but let the **observed answer pattern** determine the final decision.

- **Knowledge Mastered**

  - Answers are semantically stable with consistently low uncertainty
  - Reference: Mean accuracy $\geq 0.80$
  - A tolerance of 0.10 is allowed if reasoning remains coherent and consistently correct

- **Knowledge Missing**

  - Answers appear as scattered guesses or high-uncertainty responses
  - Reference: Mean accuracy $\leq 0.20$
  - A tolerance of 0.10 is allowed if behavior resembles random guessing

- **Knowledge Confused**

  - Answers fluctuate due to ambiguity or incomplete reasoning
  - Reference: Mean accuracy between 0.20 and 0.80
  - Mixed partial success and failure indicates fragmented knowledge

**3. Provide Explanation**
Write a concise 2–4 sentence rationale describing the model's typical response behavior and the reason for the assigned region.

### Input
Question: {question}
Gold Answer: {gold_answer}
Mean Accuracy: {mean_accuracy}
Mean Uncertainty: {mean_uncertainty}
Samples (one per line, JSON): {samples_json}

### Output Format (JSON only)

```
{
  "Knowledge_Status": "Knowledge Mastered | Knowledge Confused | Knowledge Missing",
  "Explanation": "2-4 sentence diagnostic rationale"
}
```

*Figure 9.* Prompt Template for Meta-Cognitive Region Assignment.

---

**Meta-Cognitive Search Query Prompt**

---

**Role.**You are a meta-cognitive analyst specializing in diagnosing and repairing knowledge states in large language models. Your task is to interpret an upstream diagnosis (*mastered*, *confused*, or *missing*) and produce a single targeted intervention search query.

### Objectives

1. **Extract Cognitive Tags** Identify key entities or concepts from the case that reveal what knowledge is absent, unstable, or already stable. These extracted elements are called **[Cognitive Tags]**.

2. **Formulate the Search Query** Produce ONE high-precision search query anchored on the extracted **[Cognitive Tags]** to repair cognitive distortion or expand the stable knowledge boundary.

### Global Instructions

**1. Align with Knowledge Status**

- **Knowledge Missing Strategy:** The model lacks a reliable internal representation and fails consistently. First extract **[Cognitive Tags]** by identifying the essential prerequisite entities or concepts required to reach the gold answer. Then construct a clear, definition- or concept-oriented search query anchored on these tags to retrieve the missing background knowledge needed for reconstruction.

- **Knowledge Confused Strategy:** The model produces inconsistent answers due to fragmented internal representations. First extract **[Cognitive Tags]** by identifying uncertain slots and the key entities or relations that must be clarified. Then construct a disambiguation-focused search query anchored on these tags to retrieve the missing links needed to stabilize the reasoning chain.

- **Knowledge Mastered Strategy:** The model demonstrates stable reasoning supported by a coherent internal representation. First extract **[Cognitive Tags]** by identifying the core entities or concepts underlying correct behavior. Then construct an expansion-oriented search query anchored on these tags to retrieve closely related information that extends the knowledge boundary while preserving the established structure.

**2. Search Query Requirements**

- The query must be retrieval-ready and written as a natural search instruction

- The query must be grounded in the case and anchored on the extracted **[Cognitive Tags]**

- Do not repeat the original question verbatim; reformulate it strategically

**3. Output Constraint**
You **must output exactly one JSON object**. No explanation, no markdown, no extra text.

### Input
Question: {question}
Gold Answer: {gold_answer}
Knowledge Status: {knowledge_status}
Diagnosis Explanation: {explanation}

### Output Format (JSON only)

```
{
  "Search_Query": "..."
}
```

*Figure 10.* Prompt Template for Meta-Cognitive Search Query.

---

**Knowledge Mastered Region QA Generation Prompt**

---

**Role.** You are generating QA pairs for the **Knowledge Mastered** region, where the model already demonstrates stable understanding.

### Task

Using the retrieved passages, expand around the same core fact implied by the query. Construct new QA pairs that explore related attributes, comparisons, relations, or broader background knowledge connected to this fact. Each QA pair should contain a new question grounded in the same core fact and a concise final answer (one short sentence, no explanation). Generate 1–2 QA pairs.

### Rules

1. All questions must revolve around the same core fact implied by the query.

2. Questions should extend the knowledge boundary in a controlled and relevant direction.

3. The answer must be concise (at most one short sentence, $\leq$ 20 words).

4. Do not include reasoning steps or explanations anywhere.

### Input
Query: {query}
Retrieved Passages: {retrieved_passages}

### Output Format (JSON only)

```
{
  "qa_pairs": [
    {"question": "...", "answer": "..."},
    ...
  ]
}
```

*Figure 11.* Prompt Template for QA generation in the Knowledge-Mastered region.

**Knowledge Confused Region QA Generation Prompt**

**Role.** You are generating QA pairs for the **Knowledge Confused** region, where the model shows unstable or ambiguous understanding.

### Task
Keep the original query scenario, but use the retrieved passages to repair underspecified or ambiguous parts. Construct QA pairs that clarify entities, relations, time constraints, or reasoning steps while staying close to the core factual target. Each QA pair should include a clearer and more specific question and a concise final answer (one short sentence, no explanation). Prefer generating 1 clarified main question plus 1–3 decomposed sub-questions.

### Rules

1. Preserve the original task setting, but make implicit details explicit.

2. Questions should resolve ambiguity or decompose complex reasoning into simpler parts.

3. The answer must be concise (at most one short sentence, $\leq$ 20 words).

4. Do not include reasoning steps or explanations anywhere.

### Input
Query: {query}
Retrieved Passages: {retrieved_passages}

### Output Format (JSON only)

```
{
  "qa_pairs": [
    {"question": "...", "answer": "..."},
    ...
  ]
}
```

*Figure 12.* Prompt Template for QA generation in the Knowledge-Confused region.

---

**Knowledge Missing Region QA Generation Prompt**

**Role.** You are generating training QA pairs for the **Knowledge Missing** region, where the model lacks foundational knowledge.

### Task
Using the retrieved passages as the primary source, construct QA pairs that target missing definitions or basic factual knowledge related to the query. Each QA pair should consist of a short, clear factual question and a concise final answer (one short sentence, no explanation). Focus on simple, core concepts that can be directly derived from the provided information. Generate 2–4 QA pairs.

### Rules

1. Every question must be answerable using only the retrieved passages.

2. Questions should be definition-oriented or basic fact-oriented.

3. The answer must be concise (at most one short sentence, $\leq$ 20 words).

4. Do not include reasoning steps or explanations anywhere.

### Input
Search Query: {search_query}
Retrieved Passages: {retrieved_passages}

### Output Format (JSON only)

```
{
  "qa_pairs": [
    {"question": "...", "answer": "..."},
    ...
  ]
}
```

*Figure 13.* Prompt Template for QA generation in the Knowledge-Missing region.

