# OpenReview forum: "Know More, Know Clearer: A Meta-Cognitive Framework for Knowledge Augmentation in Large Language Models"
_ICML.cc/2026/Conference — ICML 2026 spotlight_

### Official Review · Reviewer_fXoN · 2026-03-09

**Soundness:** 3
**Presentation:** 3
**Significance:** 3
**Originality:** 3
**Overall Recommendation:** 4
**Confidence:** 4

**Summary:**

This paper proposes a meta-cognitive framework for knowledge augmentation in LLMs that explicitly models the alignment between model confidence and factual correctness. The authors argue that many existing knowledge augmentation methods assume that model performance directly reflects internal knowledge, ignoring the knowledge–confidence gap that causes overconfident hallucinations or uncertain correct answers.

The framework introduces two main components:
- Cognition-Guided Knowledge Expansion (CGKE). The model partitions its knowledge space into three regions, i.e., Mastered, Confused, and Missing, based on internal uncertainty signals and sampled responses. Region-specific data augmentation strategies are then applied to expand knowledge selectively.
- Cognition-Driven Knowledge Calibration (CDKC). The model’s confidence is calibrated using entropy-based uncertainty signals and reinforcement learning (via GRPO) to align subjective confidence with objective correctness.

A key observation motivating the approach is the Structural Decay Law, which shows an exponential relationship between prediction accuracy and uncertainty, seen from Figure 1.
Experiments on 11 QA benchmarks demonstrate that the proposed framework improves both knowledge accuracy and calibration. The best configuration (two rounds of expansion and calibration) achieves significant gains over strong baselines such as RAG, CoT, and GRPO across multiple datasets.

Overall, the authors explore the key challenge of aligning model confidence with actual knowledge while expanding the model’s knowledge boundaries.

**Compliance With Llm Reviewing Policy:**

Affirmed.

**Final Justification:**

Authors' rebuttal addressed most of my concerns. I will maintain my positive score.

**Key Questions For Authors:**

- How stable are the Mastered/Confused/Missing assignments across sampling seeds?
- Does the observed uncertainty–accuracy relationship hold for larger models (e.g., 70B+)?
- Would the framework improve performance in tasks beyond QA?
- Why there are so many missing values in Table 3? Why don’t you list the Data Statistics of Training Data and Evaluation Data for Weakly-Grounded and Well-Grounded.

**Limitations:**

The authors have discussed some limitations and potential negative societal impact of their work. Except for the weaknesses, there are also some suggestions for improvement.
- A theoretical explanation would strengthen the meta-cognitive foundation of the framework.
- Testing on reasoning or planning tasks would demonstrate broader applicability.
- A sensitivity study could clarify whether performance depends heavily on this classification.

**Strengths And Weaknesses:**

## Strengths
- The paper highlights an important issue:  that LLMs often produce high-confidence incorrect answers (hallucination) or low-confidence correct answers. This mismatch between internal confidence and correctness is a known problem but is rarely addressed directly in knowledge augmentation methods.
- The idea of treating LLM training through a meta-cognitive lens (monitoring + control) is conceptually appealing. The framework operationalizes this idea through knowledge-state partitioning, uncertainty-aware augmentation, and confidence calibration via RL. This perspective is novel relative to traditional knowledge injection approaches.
- The classification of knowledge into three regions, i.e., Mastered, Confused, and Missing, is intuitive and useful for designing targeted training strategies. The region-specific augmentation strategy shown in Figure 2 clearly illustrates how the framework adapts supervision depending on the knowledge state.
- The paper provides comprehensive empirical evaluation covering 11 QA datasets, two backbone models (Qwen2.5-7B, Llama-3.1-8B) and multiple baseline paradigms. And The framework consistently improves average accuracy as shown in Table 1.
- The authors go beyond accuracy and evaluate self-knowledge ability using answerable/unanswerable questions. As shown in Figure 3, the proposed method increases correct answering on answerable questions, and improves boundary recognition for unanswerable questions. This evaluation supports the claim that the model better distinguishes known vs unknown knowledge.

## Weaknesses
- While the meta-cognitive framing is interesting, the actual techniques largely combine existing components, such as uncertainty estimation, RAG-style retrieval, supervised fine-tuning, and GRPO reinforcement learning. The contribution is mainly in integration and framing, rather than fundamentally new algorithms.
- The knowledge region classification relies on sampled outputs, uncertainty metrics, and prompt-based meta-evaluation. This process may introduce noise or instability, especially for borderline cases between "Confused" and "Missing". The paper does not analyze robustness of this classification.
- The empirical observation that accuracy decays exponentially with uncertainty is interesting, but the theoretical justification is limited, and validation across more model families would strengthen the claim.

---

> ### Author Rebuttal · Authors · 2026-03-28
>
> We greatly thank the reviewer  fXoN for the constructive feedback. We provide our detailed responses below.
>
> **Weakness 1:**
>
> We respectfully argue that our contributions extend beyond mere integration, and highlight three novel elements below.
>
> ***Discovery of the Structural Decay Law.*** To the best of our knowledge, we are the first to formalize a universal exponential decay relationship between uncertainty and accuracy across diverse LLM architectures, providing a principled foundation for operationalizing meta-cognition in LLMs.
>
> ***Meta-cognitive partitioning.*** As acknowledged by Reviewer ABAT, differentiated intervention strategies tailored to distinct knowledge regions offer a novel paradigm for knowledge augmentation and valuable reference for future research.
>
> ***Novel algorithmic contribution in CDKC.*** We propose a bidirectional entropy-based optimization algorithm within GRPO as a novel calibration mechanism. Appendix A demonstrates that vanilla GRPO amplifies conservatism bias, while our calibration loss provides the necessary escape velocity for clearer knowledge boundary awareness, validated through self-knowledge tasks.
>
> **W2:**
>
> Relying on sampled outputs to delineate knowledge boundaries is widely adopted, and LLMs have been shown to serve as reliable judges for complex cognitive distinctions. To further address this concern,  as detailed in our response to Reviewer QCeC (Issue 2), our ***human evaluation achieves a Cohen's Kappa of 0.86***, confirming substantial agreement between human judgment and the LLM's classification. We will further strengthen the robustness analysis in the revised manuscript.
>
> **W3:**
>
> We thank the reviewer for this comment. ***As presented in Appendix C.1, we verify the Structural Decay Law across six architectures spanning the Qwen, Llama, and Mistral families***, confirming it as a universal structural property of LLMs. We acknowledge that a more rigorous theoretical justification remains an important avenue for future work.
>
> **Question 1:**
>
> To validate assignment stability, we randomly sampled 1,000 instances and ran region assignment across five different random seeds, with ***95.7% of instances receiving identical classifications across all five runs***. This stability is further reinforced by our design of deriving assignments from mean statistics over K=16 independent samples, inherently mitigating random fluctuations.
>
> **Q2:**
>
> We thank the reviewer for this question. We conducted additional experiments on Qwen2.5-72B-Instruct, and the results confirm that the consistent exponential decay relationship, demonstrating that ***the Structural Decay Law generalizes effectively to larger models***. We will include this result in the revised manuscript.
>
> **Q3:**
>
> ***As detailed in Appendix C.4, we evaluate our framework on non-QA tasks including mathematical reasoning, slot-filling, dialogue, and fact-verification.*** The consistent performance improvements suggest that the proposed meta-cognitive mechanisms generalize well beyond the QA domain.
>
> **Q4:**
>
> ***All evaluation data statistics are fully reported in Table 3***. The missing entries appear exclusively in the training data section, as we only use NQ, HotpotQA, and 2WikiMQA for training, with ***all remaining datasets serving as out-of-domain test sets***. The consistent improvements across these unseen datasets demonstrate our framework's strong generalization capability.
>
> **Limitation 1:**
>
> ***As detailed in Appendix A, we provide a formal theoretical analysis grounding our framework in dynamic manifold reshaping.*** We formally analyze how standard GRPO inadvertently amplifies conservatism bias, and demonstrate that our CDKC framework provides the necessary escape velocity to overcome this limitation and synchronize subjective confidence with objective accuracy.
>
> **L2:**
>
> ***As presented in Appendix C.4, our framework consistently yields meaningful improvements on mathematical reasoning tasks.*** Given our primary focus on knowledge augmentation, reasoning and planning tasks were not included as primary targets. We recognize their value and will explore this direction in future work.
>
> **L3:**
>
> We understand the concern regarding whether performance depends heavily on the classification. Designing differentiated strategies for each cognitive region enables more fine-grained remediation of model deficiencies, inherently superior to undifferentiated approaches such as vanilla SFT or LLKD-SFT.
>
> Moreover, a substantial portion of our gains stems from knowledge boundary calibration, removing CGKE entirely still yields 42.86, significantly outperforming all baselines, ***indicating that the framework does not rely exclusively on the classification scheme***. We will incorporate a more comprehensive sensitivity analysis in the revised manuscript.
>
> We sincerely appreciate the reviewer's efforts. Due to rebuttal length constraints, if you have any remaining concerns, we warmly welcome further feedback.

---

> > ### Author Rebuttal · Reviewer_fXoN · 2026-04-02
> >
> > Dear Authors,
> >
> > Thank you for clarifying the questions I proposed and addressing most of my concerns. I will maintain my positive score.
> >
> > Best Regards,
> >
> > Reviewer fXoN

---

> > > ### Author Response · Authors · 2026-04-03
> > >
> > > Dear Reviewer fXoN,
> > >
> > > We sincerely thank you for the recognition of our work, the thorough review, and the constructive feedback, and for maintaining a positive score. We truly appreciate the time and effort invested, and will carefully address all remaining suggestions in the revised manuscript.
> > >
> > > Best regards,
> > >
> > > Authors

---

### Official Review · Reviewer_ABAT · 2026-03-10

**Soundness:** 3
**Presentation:** 3
**Significance:** 3
**Originality:** 3
**Overall Recommendation:** 5
**Confidence:** 4

**Summary:**

In this paper, the authors address the knowledge confidence gap problem by proposing a meta-cognitive framework for reliable knowledge augmentation. Specifically, the authors partition knowledge space into mastered, confused, and missing regions via internal cognitive signals, and then adapt different strategies (know more or know clearer) for cognitive misalignment correction. Experimental results verified the effectiveness of the proposed method.

**Compliance With Llm Reviewing Policy:**

Affirmed.

**Final Justification:**

The authors have addressed my concerns so I maintain my score.

**Key Questions For Authors:**

See Weaknesses above.

**Limitations:**

yes

**Strengths And Weaknesses:**

Strengths:
1. The idea of meta-cognition-based partition is novel.
2. It is interesting to adapt different strategies for different meta-cognitive region, which I think is reasonable and can be a good reference for future studies;
3. The experimental results are competitive and convincing.

Weaknesses:
1. Although I think meta-cognition-based partition is novel, it is not easy to find a valid signal or inner state for good partition. This paper uses uncertainty(the sequence-level average negative likelihood), which is ad hoc and can be further improved.

---

> ### Author Rebuttal · Authors · 2026-03-28
>
> We thank the reviewer ABAT for acknowledging the novelty of our meta-cognition-based partition. We concur that identifying a reliable internal signal for knowledge state partitioning is a non-trivial challenge. As formally established in Section 2.1 and Appendix C.1, we empirically derive the Structural Decay Law from large-scale sample distributions, demonstrating that the sequence-level average negative log-likelihood exhibits a statistically consistent exponential decay relationship with model accuracy across diverse architectures. ***This regularity substantiates that the proposed uncertainty measure captures structured information pertaining to the model's latent cognitive states, rather than constituting an arbitrary heuristic, thereby providing rigorous empirical grounding for its adoption in meta-cognitive partitioning.*** We acknowledge that relying solely on uncertainty-based signals may not yield perfect partition under all conditions, and investigating more principled internal state representations remains an important direction for future exploration.
>
> Once again, we sincerely appreciate the reviewer's recognition of our work and the valuable feedback provided.

---

> > ### Author Rebuttal · Reviewer_ABAT · 2026-04-02
> >
> > The authors have addressed my concerns so I maintain my score.

---

> > > ### Author Response · Authors · 2026-04-03
> > >
> > > Dear Reviewer ABAT,
> > >
> > > We sincerely thank you for the recognition of our work and for confirming that all concerns have been addressed. We truly appreciate the time and effort invested in reviewing our work.
> > >
> > > Best regards,
> > >
> > > Authors

---

### Official Review · Reviewer_QCeC · 2026-03-12

**Soundness:** 4
**Presentation:** 4
**Significance:** 4
**Originality:** 4
**Overall Recommendation:** 5
**Confidence:** 3

**Summary:**

In this paper, the authors propose a novel meta-cognitive framework for reliable knowledge augmentation. This framework has two main components: Cognition-Guided Knowledge Expansion (“Know More”) and Cognition-Driven Knowledge Calibration (“Know Clearer”). The authors report that this framework outperforms strong baselines consistently.

**Compliance With Llm Reviewing Policy:**

Affirmed.

**Final Justification:**

The authors addressed all of the minor issues that were raised. They supported the rebuttal with proper justifications and new results, thereby improving an already solid paper. Thus, I increased the ratings in "Soundness" and "Presentation."

**Key Questions For Authors:**

None

**Limitations:**

The paper could benefit from addressing some of the aforementioned minor issues as limitations.

**Strengths And Weaknesses:**

Soundness.
The paper is technically sound. The method is theoretically grounded. The methodology is clearly formalized with well-defined equations. The evaluation is also comprehensive.
- Minor issues:
1. The discovery of the Structural Decay Law is insightful. However, the paper does not explicitly report any goodness-of-fit metrics for different models. It is also unclear how the law handles incorrect answers. Does uncertainty decay for them, too? Investigating outliers, such as cases where high uncertainty coincides with high accuracy, could be insightful.
2. Meta-Cognitive Region Assignment is a self-diagnosis performed with an LLM. The paper provides no validation on accurate it is.
3. CGKE relies on an external retriever to fetch passages from Wikipedia. The paper does not report how sensitive CGKE is to retrieval quality.
4. The paper could benefit from human evaluation of whether the generated answers are actually more reliable or less hallucinatory.
5. The framework is complex and involves multiple moving parts, which likely leads to high computational costs. The paper could discuss whether the gains justify the overhead compared to more standard methods.

Presentation.
The submission is clearly written and well-structured. The “Related Work” section properly positions the work in the context of prior studies. The paper provides sufficient detail to facilitate reproducibility.

Significance.
This paper addresses the important issue of knowledge augmentation in LLMs. Its scope of impact is quite broad. The key insight that knowledge augmentation should be state-dependent and metacognitively guided (models need to know what they don't know before they can learn it effectively) applies to a wide variety of domains.

Originality.
This work introduces a new, theoretically grounded method that has been extensively evaluated.

---

> ### Author Rebuttal · Authors · 2026-03-28
>
> We sincerely thank the reviewer QCeC for the positive assessment and constructive comments. We hope the following clarifications address all the raised issues.
>
> **Issue 1:**
>
> ***Goodness-of-fit metrics.*** As presented in Appendix C.1, we verify the Structural Decay Law across six architectures spanning the Qwen, Llama, and Mistral families, with consistent alignment of the fitted exponential curves across all models. For example, ***the R² value for Qwen2.5-7B-Instruct reaches 0.7854***, confirming a strong fit. We will incorporate goodness-of-fit metrics for all models in the revised manuscript.
>
> ***Incorrect answers.*** The Structural Decay Law is derived from population-level statistics via multi-sample mean estimation, ***which inherently incorporates incorrect answers into the fitted distribution***. Analyzing incorrect answers individually is less straightforward, as errors stem from heterogeneous sources such as outdated knowledge, knowledge absence, each exerting a distinct influence on uncertainty. Consequently, incorrect answers alone do not readily exhibit a consistent exponential decay pattern.
>
> ***Outliers.*** We acknowledge that investigating outliers could yield valuable insights. Indeed, addressing such cases, specifically overconfident incorrect responses and underconfident correct ones, constitutes a core motivation of our framework, which aims to achieve effective calibration between accuracy and confidence. We agree that a more systematic analysis of these edge cases remains a promising direction for future exploration, and we sincerely appreciate the reviewer's valuable feedback.
>
> **Issue 2:**
>
> We thank the reviewer for this important concern. To further address this concern, we conducted a human evaluation on 300 randomly sampled instances (100 per region), where two independent annotators assessed the LLM's region assignments. ***The resulting Cohen's Kappa of 0.86 confirms substantial agreement between human judgment and the LLM's classification***, indicating that our meta-cognitive region assignment effectively captures human-perceived knowledge boundaries.
>
> **Issue 3:**
>
> We thank the reviewer for this question. The external retriever in CGKE serves as a general-purpose tool for simulating knowledge augmentation rather than a component specifically optimized for retrieval quality. Our primary focus is on validating the meta-cognitive framework itself, and the retrieval module is intentionally kept as a standard, untuned component to ensure a fair baseline. We acknowledge that higher-quality retrieval would provide richer external knowledge, thereby further amplifying the benefits of our framework.
>
> **Issue 4:**
>
> We thank the reviewer for this valuable suggestion. We agree that human evaluation of answer reliability and hallucination reduction would provide a more comprehensive assessment of our framework's practical effectiveness. We will incorporate human evaluation as an additional evaluation dimension in the revised manuscript to further strengthen the overall assessment.
>
> **Issue 5:**
>
> We thank the reviewer for this concern. Taking Qwen2.5-7B-Instruct as an example, we address the computational cost from two perspectives.
>
> ***Knowledge Expansion (CGKE).*** The additional sampling and cognitive region partitioning in CGKE introduce extra overhead. However, this stage demonstrates that fine-grained, region-aware knowledge augmentation yields superior performance over undifferentiated fine-tuning, a finding we believe offers valuable insights for future research, as also acknowledged by Reviewer ABAT.
>
> ***Knowledge Calibration (CDKC).*** The majority of our performance gains stem from CDKC, which introduces a calibration loss on top of GRPO without significant extra overhead. In fact, compared to methods that require constructing additional training data, our approach requires no additional data generation cost. As shown below, ***Compared to standard GRPO, CDKC adds only 1 hour of training time (35h → 36h) while achieving a meaningful performance gain (43.91 → 44.97)***, demonstrating a highly favorable cost-performance trade-off.
>
> | Method    | Data Generation Time | Training Time | AVG ACC |
> | --------- | -------------------- | ------------- | ------- |
> | Know what | 13h                  | 0.8h          | 28.88   |
> | CRew-DPO  | 13h                  | 1.5h          | 30.26   |
> | BARREL    | 11h                  | 35h           | 33.04   |
> | GRPO      | -                    | 35h           | 43.91   |
> | CDKC      | -                    | 36h           | 44.97   |
>
> Furthermore, for those concerned about the overall computational cost of the full framework, our ablation study demonstrates that removing the CGKE stage entirely still yields an average score of 42.86, significantly outperforming other baselines, ***suggesting that even the lightweight version of our framework delivers substantial gains at reduced computational cost***.

---

> > ### Author Rebuttal · Reviewer_QCeC · 2026-04-02
> >
> > Thank you for your exhaustive response. I consider all issues resolved.

---

> > > ### Author Response · Authors · 2026-04-03
> > >
> > > Dear Reviewer QCeC,
> > >
> > > We sincerely thank you for the recognition of our work and for confirming that all issues have been resolved. We truly appreciate the time and effort invested, and will carefully incorporate all suggestions into the revised manuscript.
> > >
> > > Best regards,
> > >
> > > Authors

---

### Official Review · Reviewer_ZkRh · 2026-03-13

**Soundness:** 4
**Presentation:** 4
**Significance:** 3
**Originality:** 3
**Overall Recommendation:** 5
**Confidence:** 3

**Summary:**

This study proposes a novel meta-cognitive framework that expands a model’s knowledge and calibrates the gap between its internal confidence and actual performance, based on LLMs' internal uncertainty signal (the Structural Decay Law). It consists of two components: CGKE (Cognition-Guided Knowledge Expansion) and CDKC (Cognition-Driven Knowledge Calibration). The framework alternates between these two components, CGKE splits given training instances into three regions (mastered/confused/missing) and expands knowledge via Supervised Fine-Tuning, and CDKC improves calibration using Group Relative Policy Optimization (GRPO) approach through different strategies for each region. The authors conduct extensive evaluations on a total of 11 popular commonsense and multi-hop QA benchmarks, demonstrating both performance improvements and better alignment between the models' internal knowledge and objective correctness through meta-cognitive assessment.

**Compliance With Llm Reviewing Policy:**

Affirmed.

**Final Justification:**

The authors addressed my questions about their experimental results and design choices through additional experiments and  qualitative validation. Further ablation studies would be helpful, but the current results sufficiently support the paper. Therefore, I increased the soundness score.

**Key Questions For Authors:**

- Q1) Although Section 5.2 presents an ablation study that removes each knowledge region, it remains unclear whether each strategy is indeed best suited to its corresponding region, at least among the three strategies considered. For example, the query-generation prompts could be swapped across regions and evaluated across all nine possible combinations to assess their suitability for the target region.
- Q2) The reward coefficients $\lambda_1$ and $\lambda_2$ in Eq.10 are important hyperparameters and both set to 0.001. I wonder whether these values may be too small to have a meaningful effect during training. It would be helpful to clarify how these values were selected.
- Q3) In the ablation study (Table 2), CGKE significantly underperforms CDKC w/o CGKE. it is surprising, considering the role of CDKC is knowledge calibration. It would be helpful if the authors could provide some intuition or discussion on why this result arises.

**Limitations:**

See weaknesses and questions.

**Strengths And Weaknesses:**

Strengths
- The credibility of LLMs and the calibration of their knowledge and confidence are timely and important topics.
- A comprehensive and scalable framework that integrates knowledge augmentation and calibration, with a tailored approach to handling different states of LLM knowledge.
- The empirical analysis of the Structural Decay Law supports its use as an uncertainty signal throughout the paper.
- The authors conduct extensive experiments and analyses, providing broad empirical evidence for the effectiveness of the proposed framework.
- The writing and presentation are clear and easy to follow.

Weaknesses
- W1) The proposed framework heavily relies on LLMs to classify questions into the three knowledge categories and to generate queries for each category. However, the verification of these LLM-generated outputs is limited. Conducting human verification on the tranining samples generated by LLMs and a case study on their differences for each region is highly recommended. Although the framework is based on self-reflection, such analysis is still important to show that it works as intended by properly capturing the differences among the regions.
- W2) The paper lacks qualitative analysis and demonstrantion of whether the generated queries meaningfully reflect the characteristics of each knowledge region. Having some examples of generated queries and emphasizing the differences between generated queries for each region would be helpful.
- W3) Section 5.3 addresses the evaluation of meta-cognitive and calibration but relies heavily on the appendix for details. It would be helpful to move Appendix C.5. to the main text to emphasize the contribution of CDKC.


suggestion: figures containing a large number of dots (e.g., Figures 1 and 5, as well as several figures in the appendix) may be too heavy to load or print in some environments. I recommend converting them to images.

---

> ### Author Rebuttal · Authors · 2026-03-28
>
> We greatly thank the reviewer ZkRh for insightful comments. We provide our responses to the comments as follows.
>
> **W1) & W2):**
>
> We fully agree that human verification and case studies are important for validating the framework's design intent.
>
> ***Human Verification.*** During development, we conducted manual sampling verification and iteratively refined the generation prompts. All training samples are grounded in externally retrieved passages as an additional quality guarantee, consistent with Self-Instruct, which demonstrates that LLM-generated data can yield substantial gains even under imperfect conditions. We will elaborate further in the revised manuscript.
>
> ***Case Study across Regions.*** To further illustrate that the generated queries meaningfully reflect the characteristics of each knowledge region, we present representative examples below.
>
> - ***Missing.*** Question: "Who played the Oompa Loompas in Charlie and the Chocolate Factory?"(answer: Deep Roy). The model consistently fails to produce the correct answer. Generated sample: Q: How many Oompa-Loompas did Deep Roy play? A: 165.
> - ***Confused.*** Question: "Which magazine was started first, Arthur's Magazine or First for Women?"(answer: Arthur's Magazine). The model produces inconsistent answers across multiple samples. Generated samples: Q1: When was Arthur's Magazine started? A1: 1844; Q2: When was First for Women started? A2: 1989.
> - ***Mastered.*** Question: "Who starred in Deepwater Horizon?"(answer: Kurt Russell). The model already answers correctly with high confidence. Generated sample: Q: What was Kurt Russell's role in Deepwater Horizon? A: A BP representative.
>
> These examples confirm that the generated queries exhibit meaningfully distinct characteristics across the three regions, ***each precisely aligned with the specific knowledge deficiency of the corresponding region, introducing missing knowledge structures, resolving conflicting facts, and expanding existing knowledge boundaries respectively***. This validates that the framework operates as intended. We will provide a more comprehensive qualitative analysis in the revised manuscript.
>
> **W3) & Suggestion:**
>
> We thank the reviewer for these helpful suggestions. In the revised manuscript, we will incorporate Appendix C.5 into the main text to better highlight the contribution of CDKC, and convert the dot-heavy figures to rasterized images to ensure smooth loading and printing.
>
> **Q1):**
>
> We sincerely appreciate the reviewer's feedback. Due to rebuttal length and time constraints, completing all nine combinations is difficult at present. We argue that the three strategies are designed with fundamentally distinct objectives: introducing new knowledge structures (Missing), disambiguating conflicting knowledge (Confused), and expanding knowledge boundaries (Mastered). Swapping these strategies across regions would be semantically misaligned with their design intent. As illustrated in W1), ***applying a boundary-expansion strategy to a knowledge missing region would clearly fail to address its fundamental deficiency***. We will incorporate a systematic cross-region prompt suitability analysis in the revised manuscript.
>
> **Q2):**
>
> ***The value λ₁=0.001 follows the standard KL coefficient setting in DeepSeek-R1. For λ₂, rather than treating it as an independent hyperparameter to tune, we set it equal to λ₁ based on the theoretical analysis in Appendix A***, where the training process is formulated as dynamic manifold reshaping driven by a composite vector field. The KL and cognitive calibration constraints compete and cooperate jointly, and setting them to the same scale prevents either from dominating, ensuring balanced optimization.
>
> To address this concern that 0.001 may be too small, we provide a sensitivity analysis over λ₂ on Qwen2.5-7B-Instruct below. When λ₂=0 (standard GRPO), average performance drops by 1.06 points, ***demonstrating a clear and measurable impact.*** As long as λ₂ does not deviate significantly from λ₁, the framework consistently outperforms GRPO, confirming its robustness. Beyond performance, introducing λ₂ enables better-calibrated self-awareness of knowledge boundaries, a core objective of our framework. We will supplement a more comprehensive sensitivity analysis in the revised manuscript.
>
> |  λ₂   | AVG ACC |
> | :---: | :-----: |
> |   0   |  43.91  |
> | 0.001 |  44.97  |
> | 0.003 |  44.49  |
> | 0.005 |  45.04  |
> | 0.007 |  44.69  |
> | 0.01  |  43.84  |
>
> **Q3):**
>
> We thank the reviewer for this observation. Although CGKE enriches the model's knowledge, expanded knowledge alone does not resolve the misalignment between confidence and accuracy. ***Without calibration, the model may still be uncertain about correct answers or overconfident about incorrect ones, preventing acquired knowledge from being reliably expressed.*** CDKC addresses this gap directly, explaining why calibration alone yields stronger gains than knowledge expansion alone.

---

> > ### Author Rebuttal · Reviewer_ZkRh · 2026-04-02
> >
> > I appreciate the authors' response. My questions have been resolved, and I understand the time constraints regarding the experiment I suggested. I have updated my score to 5.

---

> > > ### Author Response · Authors · 2026-04-03
> > >
> > > Dear Reviewer ZkRh,
> > >
> > > We sincerely thank you for the recognition of our work, the positive feedback, and for updating the score. We truly appreciate the time and effort invested, and will carefully incorporate all suggestions into the revised manuscript.
> > >
> > > Best regards,
> > >
> > > Authors

---

### Decision · Program_Chairs · 2026-04-30

**Decision:**

Accept (spotlight)

**Comment:**

All four reviewers recommend acceptance (5/5/5/4), recognizing the novelty of the meta-cognitive partitioning idea, comprehensive evaluation, and clear presentation.

The framework's dual design of knowledge expansion and confidence calibration was considered a meaningful contribution to the field. Reviewers raised concerns about human verification of region assignments, computational overhead, and robustness of the uncertainty signal, all of which were adequately addressed during rebuttal with additional experiments and analysis. The authors should incorporate the promised qualitative case studies, human evaluation details, and calibration analysis into the camera-ready main text.

I recommend acceptance.